



# A Model for Urban Biogenic CO₂ Fluxes: Solar-Induced Fluorescence for Modeling Urban biogenic Fluxes (SMUrF v1)

Dien Wu[1,*], John C. Lin[1], Henrique F. Duarte[1,‡], Vineet Yadav[2], Nicholas C. Parazoo[2], Tomohiro Oda[3,4,5], and Eric A. Kort[6]

[1]Department of Atmospheric Sciences, University of Utah, Salt Lake City, USA
[2]NASA Jet Propulsion Laboratory, California Institute of Technology, Pasadena, USA
[3]Goddard Earth Sciences Technology and Research, Universities Space Research Association, Columbia, USA
[4]Global Modeling and Assimilation Office, NASA Goddard Space Flight Center, Greenbelt, USA
[5]Department of Atmospheric and Oceanic Science, University of Maryland, College Park, USA
[6]Climate and Space Sciences and Engineering, University of Michigan, Ann Arbor, USA
[*]Now at Division of Geological and Planetary Sciences, California Institute of Technology, Pasadena, USA
[‡]Now at Earth System Science Center, National Institute for Space Research, São José dos Campos, Brazil

*Correspondence to*: Dien Wu (dienwu@caltech.edu)

**Abstract.** When estimating fossil fuel carbon dioxide (FFCO₂) emissions from observed CO₂ concentrations, the accuracy can be hampered by biogenic carbon exchanges during the growing season even for urban areas where strong fossil fuel emissions are found. While biogenic carbon fluxes have been studied extensively across natural vegetation types, biogenic carbon fluxes within an urban area have been challenging to quantify due to limited observations and differences between urban versus rural regions. Here we developed a simple model representation, i.e., Solar-Induced Fluorescence (SIF) for Modeling Urban biogenic Fluxes ("SMUrF"), that estimates the gross primary production (GPP) and ecosystem respiration ($R_{eco}$) over cities around the globe. Specifically, we leveraged space-based SIF, machine learning, eddy-covariance flux data, and additional remote sensing-based products, and developed algorithms to gap fill fluxes for urban areas. Grid-level hourly mean net ecosystem exchange (NEE) are extracted from SMUrF and evaluated against 1) non-gapfilled measurements at 67 eddy-covariance (EC) sites from FLUXNET during 2010 – 2014 (r > 0.7 for most data-rich biomes), 2) independent observations at two urban vegetation and two crop EC sites over Indianapolis from Aug 2017 to Dec 2018 (r = 0.75), and 3) an urban biospheric model based on fine-grained land cover classification within Los Angeles (r = 0.83). Moreover, we compared SMUrF-based NEE with inventory-based FFCO₂ emissions over 40 cities and addressed the urban-rural contrast regarding both the magnitude and timing of CO₂ fluxes. By examining a few summertime satellite tracks over four cities, we found that the urban-rural gradient in column CO₂ (XCO₂) anomalies due to NEE can sometimes reach ~0.5 ppm and be close to XCO₂ enhancements due to FFCO₂ emissions. With rapid advances in space-based measurements and increased sampling of SIF and CO₂ measurements over urban areas, SMUrF can be useful for informing the biogenic CO₂ fluxes over highly vegetated regions during the growing season.





# 1 Introduction

Climate change and urbanization are two major worldwide phenomena in recent decades. In close connection with both themes, cities have attracted increasing attention from both researchers and policymakers. Urban ecosystems can be unique and complex compared to natural ecosystems because of the wide variety of land use/land covers in cities, along with relatively

higher levels of $CO_2$ concentrations and air temperatures than surrounding rural ecosystems (Wang et al., 2019). The consequences of climate change, such as severe heat, drought, and water shortage events, may be exacerbated particularly over (semi)arid and/or developing cities (Nowak and Crane, 2002; Rosenzweig et al., 2015), which result in possible population movement from increasingly hot/dry places to relatively cool/moist ones. Meanwhile, rapid urban expansion and population growth may contribute to the rise in the total anthropogenic $CO_2$ emissions into the atmosphere. Human activities have been

continuously modifying the urban and natural vegetation and soil, e.g., expansion of agricultural lands at the cost of the natural landscape, leading to less reversible ecological and climatic impacts (Ellis and Ramankutty, 2008; Hutyra et al., 2014; Pataki et al., 2006). Hence, urban areas function as both biophysical and socioeconomic systems, and studying their carbon sources/sinks facilitates understanding cities' roles in the global carbon cycle.

To study the urban carbon pool and its exchange with the atmosphere, the top-down approach based on measured concentrations of atmospheric $CO_2$ is commonly used. McRae and Graedel (1979) noted over four decades ago that separation between anthropogenic and biogenic $CO_2$ flux signals is needed to interpret urban $CO_2$ observations. Biogenic $CO_2$ fluxes are found to modify surface and even atmospheric column concentrations of downwind $CO_2$ (e.g., Lin et al., 2004, Turnbull et al., 2015, Hardiman et al., 2017, Sargent et al., 2018, Ye et al., 2020). Simulated biogenic contributions over the Pearl River Delta,

China, result in column $CO_2$ ($XCO_2$) anomalies of nearly 0.5 ppm in the summer, which can be non-negligible compared to $FFCO_2$ signals ranging from 1 to 2 ppm (Ye et al., 2020). Thus, assessing the contributions from biogenic fluxes in the observed signals is crucial for inversely quantifying $FFCO_2$ emissions for cities and yet challenging especially given limited urban fluxes observations across the globe. Although deciduous trees are found to be the dominant tree type over 328 global cities (Yang et al., 2015a), a more accurate approximation of the vegetation coverages, types, and biological activities in cities is hard to

obtain.

Existing approaches to separate biogenic and anthropogenic $CO_2$ components involve the use of ancillary tracers and terrestrial biosphere models. For instance, since radiocarbon ($^{14}C$) has decayed in fossil fuels, $^{14}C$ serves as a tracer for the combustion of FF emission (Turnbull et al., 2015). Carbonyl sulfide (COS) shares a similar seasonal variation as $CO_2$ over the land, a result

of biospheric sinks (Kettle et al., 2002). However, measurements of $^{14}C$, COS, and $CO_2$ fluxes are fairly limited over cities around the globe. Besides observations, many global terrestrial biosphere models provide insights for informing and constraining $CO_2$ fluxes at continental to global scales (Huntzinger et al., 2013; Knorr and Heimann, 2001; Philip et al., 2019), but their relatively coarse resolution and simplifications on urban biosphere limits the use for studying urban carbon cycles. In





addition, only a select few among biospheric models are designed for simulating urban biogenic fluxes. Research has revealed urban-rural differences in vegetation and soil properties, in part due to management strategies and environmental conditions, which complicate the flux quantification (Decina et al., 2016; Hardiman et al., 2017; Smith et al., 2019; Vasenev and Kuzyakov, 2018). Among these few models, the Urban Vegetation Photosynthesis and Respiration Model (urbanVPRM, Hardiman et al., 2017) is an empirical model that incorporates the urban heat island effect and impervious surface area into its flux calculations and currently uses conventional greenness indices e.g., the Enhanced Vegetation Index (EVI).

Our work is primarily motivated by the relatively coarse spatial grid spacing and the lack of account or simplifications of urban ecosystems in most models. We attempted to bridge between coarse-scale global biospheric models and highly customized local models to offer a global solution to modeling biogenic $CO_2$ fluxes within and around urban areas, which would provide insight into $CO_2$ partitioning between fossil fuel and biogenic components.

Thanks to advances in spaceborne and ground-based measurements, solar-induced fluorescence (SIF) has been retrieved successfully from various satellites and has proven to be an effective proxy for photosynthesis and thus modeling gross primary production (GPP) (Frankenberg et al., 2011; Guanter et al., 2014; Joiner et al., 2013; Yang et al., 2015b). Even though conventional greenness indices like EVI adopted in many models are often highly resolved in space, they mainly indicate photosynthesis capacity while SIF is more closely related to the actual photosynthesis activity (Luus et al., 2017). Consequently, SIF better tracks the seasonal and interannual variations in GPP for various plant functional types (Luus et al., 2017; Smith et al., 2018; Turner et al., 2020; Zuromski et al., 2018) and their responses to physiological stress (Magney et al., 2019). Several spatially continuous SIF products have been produced, beyond the limited satellite soundings where SIF are being retrieved (Li and Xiao, 2019a; Turner et al., 2020; Zhang et al., 2018). Moreover, linear correlations between GPP and SIF has been derived at the daily and landscape scale (Magney et al., 2019; Sun et al., 2018; Turner et al., 2020; Zhang et al., 2018; Zuromski et al., 2018) and used to generate spatial maps of GPP fluxes (Li and Xiao, 2019b; Yin et al., 2020). SIF information has also been incorporated into existing process-based biospheric models and data assimilation systems (MacBean et al., 2018; Van Der Tol et al., 2009). Within the context of the urban biosphere, SIF is capable of revealing the urban-rural gradient in photosynthetic phenology (Wang et al., 2019). Given all these advantages, SIF would potentially benefit the GPP estimates and $CO_2$ fluxes partitioning over cities.

Ecosystem respiration ($R_{eco}$), the other component of the net ecosystem exchange (NEE), is defined as the sum of the autotrophic ($R_A$) and heterotrophic ($R_H$) components. In terms of modeling urban $R_{eco}$, urbanVPRM follows the conventional approach of VPRM (Mahadevan et al., 2008) to estimate an initial air temperature-scaled $R_{eco}$ and splits $R_{eco}$ into equal components ($R_A$ and $R_H$), which are then modified by incorporating impervious fractions and urban heat island effects (Hardiman et al., 2017). However, the exact partitioning of $R_{eco}$ between $R_A$ versus $R_H$ or the above- versus below-ground respiration can be challenging and highly uncertain, as acknowledged in Hardiman et al. (2017), and the initial $R_{eco}$ that





urbanVPRM modified may be an overly simplistic function of ambient air temperature. After all, the complexity of biological and non-biological processes of $R_{eco}$ and the lack of mechanistic understanding of how biotic and abiotic factors affect $R_{eco}$ make the mechanistic modeling quite challenging.

Given the complexity in ecosystem modeling, machine learning (ML) techniques have been increasingly applied to help answer complicated, entangled problems through extracting patterns from data streams for predictions and generalizations. Reichstein et al. (2019) provided a fairly comprehensive review on the many applications of ML techniques in solving geoscience and remote sensing problems and identified challenges in successfully adopting ML approaches— e.g., interpretability, integration with physical understanding and modeling, and the ability to cope with model/data
uncertainties. For instance, artificial neural network (NN) has been used to generate SIF beyond satellite soundings (Li and Xiao, 2019a; Zhang et al., 2018), harmonize multiple SIF satellite instruments (Wen et al., 2020), map carbon and energy fluxes (Tramontana et al., 2016), and predict and reveal the trend in global soil respiration (Zhao et al., 2017).

In this paper, we present a model representation of GPP, $R_{eco}$, and NEE fluxes targeting urban areas around the globe, the
*Solar-Induced Fluorescence (SIF) for Modeling Urban biogenic Fluxes* ("SMUrF"), by taking advantages of SIF and NN technique. Our main objectives include: 1) examine the biogenic and anthropogenic $CO_2$ fluxes and their temporal variations over urban and rural areas; and 2) demonstrate one application of SMUrF to help interpret satellite $CO_2$ observations by revealing the urban-rural gradient in biogenic $CO_2$ fluxes along satellite swaths of the Orbiting Carbon Observatory 2 (OCO-2, Crisp et al., 2012).

**2 Data and methodology**

SMUrF incorporates SIF as an indicator of photosynthesis, along with possible drivers for $R_{eco}$ from air and soil temperatures and SIF-driven GPP, and performs hourly downscaling using reanalysis-based temperature and radiation fields (**Fig. 1**). SMUrF simulates hourly biogenic fluxes at 0.05°. Gridded uncertainties of daily mean fluxes are quantified by assigning biome-specific coefficient of variations (CVs) from model-data comparisons (**Sect. 2.5**). To translate the anthropogenic and
biogenic contributions in column $CO_2$ concentration space from modeled fluxes, we adopted an atmospheric transport model (**Sect. 2.6**).

**2.1 Input datasets**

Similar to many terrestrial biospheric models, SMUrF estimates gridded GPP, $R_{eco}$, and NEE ($R_{eco}$ – GPP) fluxes based on biomes. The main required data streams are summarized in a table in **Fig. 1**, including 1) 500 m MODIS-based annual land
cover classification; 2) 0.05° spatiotemporally contiguous SIF product; 3) above-ground biomass (AGB) from GlobBiomass; 4) eddy-covariance (EC) flux measurements; and 5) gridded products of air and soil temperatures.



### 2.1.1 Land cover classification

Sophisticated land cover classifications (e.g., NLCD 2016) are often available for limited regions. Thus, the land cover types defined by the International Geosphere–Biosphere Programme (IGBP) from MCD12Q1 are adopted to inform biomes over global lands. We accessed the preprocessed version of MCD12Q1 v006 (Friedl and Sulla-Menashe, 2019) with latitude and

longitude grids from **App**lication for **E**xtracting and **E**xploring **A**nalysis **R**eady **S**amples. Twelve biomes include croplands (CRO), closed and open shrublands (CSHR, OSHR), deciduous/evergreen broadleaf/needleleaf forests (DBF, DNF, EBF, ENF), grasslands (GRA), mixed forests (MF), savannas (SAV), woody savannas (WSAV), and permanent wetlands (WET). Unfortunately, MCD12Q1 simply treats the entire urban area as one category (URB), while in reality both grass and trees exist within the MODIS-based URB category. Therefore, we developed an algorithm to approximate the vegetation types and

fractions in cities (**Sect 2.2.2**).

### 2.1.2 Data for GPP estimates

Since current satellites like OCO-2 retrieve SIF at discrete timesteps and with spatial coverage limited along a swath, a spatiotemporally Contiguous SIF (CSIF, Zhang et al. 2018) product together with GPP fluxes from FLUXNET2015 (Pastorello et al., 2017) were used to calculate biome-specific GPP-CSIF slopes ($\alpha$, **Supplementary Fig. S1**). Over 80 global EC tower

sites with screened data points (quality flag < 3) from 2010 to 2014 are chosen to represent various biomes. CSIF offers global 4-day mean SIF at the grid spacing of 0.05° during 2000–2018 using the NN approach (Zhang et al., 2018). The NN model in CSIF is constructed based upon OCO-2 SIF and four broadband reflectances from MCD43C4 V006 under clear-sky conditions and is used for mapping SIF beyond sounding locations. CSIF agrees well with SIF retrievals from OCO-2 and GOME-2, considering the inevitable spatial mismatch between CSIF (0.05°) and the direct sounding-level SIF measurements (OCO-2's

footprint of ~1 km × 2 km). The two largest biases of CSIF with respect to OCO-2 SIF arise from croplands (–12.72 %) and urban areas (–14.59 %), caused by the saturation effect in broadband reflectances and built-up contaminations to the reflectance signal, respectively (Zhang et al., 2018). High chlorophyll concentrations in croplands may not be fully captured by the reflectance data used for training (Zhang et al., 2018). To compensate for the potential bias of urban CSIF, we simply scale up the GPP-SIF slope for urban areas (more in **Sect. 2.2.2**). In addition, we compared the clear-sky instantaneous CSIF with

TROPOMI-based downscaled SIF and vegetated fractions inferred from the WUDAPT product (**Supplementary Figs. S2-S3**). These comparisons confirmed CSIF's overall performance and capability in revealing urban-rural gradient in biogenic activities. Please refer to **Appendix A** for details.

### 2.1.3 Data for predicting urban vegetation globally

To assign trees and grass in the MODIS-based urban domain, several steps are carried out to approximate the relative fractions

of 1) tree versus non-tree, 2) individual tree types, and 3) grassland versus shrubland. Relative tree fractions (i.e., ratio of total tree fractions to total vegetated fractions) can be extracted from two possible data sources—i.e., a 0.6 m urban land cover





product (Coleman et al. 2020) and a 250 m Vegetation Continuous Fields (VCF) from MOD44B (Dimiceli et al., 2015). Both products offer gridded tree and vegetated fractions. The former one is produced only over Los Angeles via random forest algorithms by trained on Sentinel-2 (~5 m) and NAIP (~0.6 m) optical imagery (Coleman et al., 2020) and possesses much higher tree fractions than VCF (see comparisons in **Sect. 2.2.2**). Thus, we decided not to utilize MODIS VCF for indicating
urban vegetation in this work, only for comparisons with tree fractions derived from Coleman et al. (2020).

To better approximate relative tree fractions over cities worldwide, we instead grab 100 m AGB from GlobBiomass dataset (Santoro et al. 2018) as spatial proxy (see methodological explanations in **Sect. 2.2.2**). GlobBiomass deployed a complex retrieval algorithm system which involves a series of retrieval algorithms using the radar backscatter and several other data
such as Laser measurement from ICESAT (Schutz et al., 2005), the tree and land cover data (e.g., from Landsat), and collections of reanalysis and models. AGB and its grid-level uncertainty have a unit of tons ha$^{-1}$, by definition describe the "oven-dry weight of the woody parts (stem, bark, branches, and twigs) of all living trees excluding stump and roots" (Santoro et al. 2018). GlobBiomass AGB has demonstrated good agreement against independent data products for different continents (Santoro et al. 2018).

**2.1.4 Reanalyses for training and predicting R$_{eco}$**

In efforts to train and predict R$_{eco}$ via neural network models, we chose GPP, air and soil temperatures (T$_{air}$ and T$_{soil}$) as explanatory variables (see descriptions of R$_{eco}$ estimates in **Sect. 2.3**). Modeled T$_{air}$ and T$_{soil}$ are grabbed from the ECMWF ReAnalysis-5 (ERA5, 0.25°; Copernicus Climate Change Service Information [2017]) for the entire globe or from Daymet (1km; Thornton et al., 2016) and the North American Land Data Assimilation System (NLDAS, 0.125°; Xia et al. 2012) as
alternative inputs for CONUS runs (**Fig. 1**). It is worth pointing out that different models/reanalysis provide T$_{air}$ at 2 meters above ground but T$_{soil}$ at different soil depths. For instance, four soil depths from NLDAS range from 10, 30, 60, and 100 cm below ground, whereas ERA5 simulates mean T$_{soil}$ over four vertical layers, i.e., $0-7$, $7-28$, $28-100$, and $100-289$ cm. Measured soil depths from FLUXNET are even more complicated and vary among sites with the most common shallowest soil depth being ~2 cm below ground. To reconcile differences in soil depths, we chose measured T$_{soil}$ from the shallowest
layer in both the model and observational datasets and separately build NN models (**Sect. 2.3**).

**2.1.5 Data products for flux comparisons**

Observations of hourly NEE fluxes from FLUXNET2015 and the Indianapolis Flux Experiment (INFLUX, Davis et al., 2017; Wu et al., 2020a) along with model outputs from urbanVPRM over Los Angeles are used for flux comparisons (**Sect. 3.2**). We only used non-gapfilled measured NEE fluxes from FLUXNET2015 (with quality flag of 0) for validation. And since NEE
fluxes measured from INFLUX have not been gap-filled, we resampled hourly modeled fluxes according to the hour intervals that are available in these flux measurements.



The INFLUX project includes EC flux measurements which accompany the tower- and aircraft-based GHG mole fraction measurements. These sites have been periodically moved to sample different components of the urban landscape. For the period from Aug 10, 2017 to June 7, 2019, these flux towers were deployed at two urban vegetation sites (#1 and #4) and two agricultural sites (#2 and #3). Because CSIF is not available beyond 2018 as of this writing, we cannot yet extend the flux

comparison into 2019. Soybean was primarily grown around site #2 and only corn was grown around site #3 in 2018. Two urban vegetation sites are located in a cemetery area (site #1) and a golf course (site #4). Fluxes were computed using EddyPro software (LI-Cor, Inc., Lincoln, NE; Biosciences, 2012, 2017) and then post-processed to filter out data when a) the LI-COR gas analyser signal strength was low; and b) during periods of weak turbulence (Wu et al., 2020a). It is worth noting that the INFLUX network provides independent evaluations since the network influces flux sites in urban areas, which are lacking

from FLUXNET2015.

The urbanVPRM model, applied over Los Angeles here, estimates GPP from a light use efficiency modeling perspective driven by reanalysis-based Photosynthetically Active Radiation (PAR) and satellite-derived EVI and Land Surface Water Index for phenology and water availability; and estimates $R_{eco}$ via an air temperature function with extra modifications to air temperature

due to urban heat island effect and impervious surface area. These modeled fluxes rely on a high-resolution land cover classification available at 60 cm over Los Angeles (Coleman et al., 2020; **Sect. 2.1.3**). Due to differences in grid spacing between models, we aggregated and re-projected the monthly urbanVPRM fluxes from 30 m to 0.05° to match SMUrF for purposes of comparison.

### 2.2 GPP estimates

We used 4-day mean clear-sky SIF from CSIF and 4-day mean observed GPP fluxes from 89 global eddy-covariance (EC) towers from FLUXNET2015 to derive biome-specific GPP-SIF slopes ($\alpha$, **Supplementary Fig. S1**) with special treatments to $\alpha$ over croplands and urban areas described in the following subsections. Our derived $\alpha$ values are in proximity to those reported in Zhang et al. (2018) from 40 towers and are then assigned to each 500 m grid cell in SMUrF according to its corresponding biome type.

### 2.2.1 C3/C4 partitioning of croplands

GPP-SIF relationships differ between C4 and C3 crops at the canopy scale, since GPP for C3 crops may saturate at high PAR levels. Therefore, different $\alpha$ are suggested in use for C3 versus C4 crops (He et al., 2020; Yin et al., 2020). Despite the focus of SMUrF on urban areas, we still differentiate C4 from C3 crops and estimate two different $\alpha$ values for EC sites dominated by either C3 or C4 crops. Next, the Spatial Production Allocation Model (SPAM 2010V1.1, You et al., 2014) is used for areal

estimates of 42 crop species, among which the following are identified as C4 crops: maize, pearl and small millet, sorghum, and sugarcane. As a result, we generated maps of relative C3/C4 ratios at a grid spacing of 10 km for the entire world (upper two panels in **Fig. 2**). Four of the selected 13 cropland EC FLUXNET sites fall within grid cells with high C4 ratio of > 50%;



the remaining sites fall into grid cells with C4 ratios of < 10%. We thereby estimate $\alpha_{C4}$ of ~35.6 [$\mu$mol m$^{-2}$ s$^{-1}$]:[mW m$^{-2}$ nm$^{-1}$ sr$^{-1}$] based on sites with high C4 ratio and $\alpha_{C3}$ of ~19.7 [$\mu$mol m$^{-2}$ s$^{-1}$]:[mW m$^{-2}$ nm$^{-1}$ sr$^{-1}$] from the rest 9 cropland sites. Eventually, we calculated the weighted mean $\alpha$ according to the spatial C3-C4 ratio and identified tropical regions, mid-west US, northeastern China, and spots in India and south Africa as regions with higher $\alpha$ and C4 crop ratios (bottom panel in **Fig.**

**2**). These weighted mean $\alpha$ will only be activated over MODIS-based croplands, as informed by MODIS.

### 2.2.2 Modification to urban vegetation

We next turn to the estimate of $\alpha$ over MODIS-based "urban" category shown in the following three steps (**Fig. 1**):

1) *Estimate the relative tree cover fraction (tree_rel = tree / vegetated).* A power-law relationship (purple equations and lines in **Fig. 3a**) between AGB bins and relative tree fractions calculated from NAIP-based land cover classification is
used to predict relative tree fractions (bottom panel in **Fig. 3b**). Although the binning procedure may not fully recreate the variations in tree fractions especially for grid cells with zero AGB (dark red hexagons in **Fig. 3a**), the AGB-based relative tree fraction is associated with a smaller deviation (bias of +2.3 %) than those in MOD44B (–23.5 %), when the high-res NAIP-based product is compared (**Fig. 3b**).

2) *Estimate relative fractions of five tree types, grass, and shrub based on climatology.* The relative tree fractions will further be divided into five specific tree types (i.e., DBF, DNF, EBF, ENF, MF) and the relative non-tree vegetated fractions (i.e., 1 – *tree_rel*) will be equally split into half grass and half shrub (2$^{nd}$ row in **Fig. 4**). Specifically, we extract the absolute fraction of each tree type from MCD12Q1 (**Supplementary Fig. S4a**) and predict the relative fraction as functions of latitudes—i.e., high fractions of ENF over high-latitudes, EBF over tropical lands, and DBF plus MF over the midlatitudes
(**Supplementary Fig. S4b**).

3) *Calculate weighted mean α and scale up α for urban grids.* α for urban areas are weighted mean values calculated from biome-specific $\alpha$ values and their corresponding biome fractions (in step 2). To account for the potential underestimation in CSIF over cities of about –14.5 % (Zhang et al., 2018), we scaled up urban $\alpha$ by 1.145.

In the end, $\alpha$ at 500 m over urban and natural lands (3$^{rd}$ row in **Fig. 4**) are aggregated to 0.05° and multiplied by CSIF to arrive at GPP at 0.05°. The exact partitioning between grass and shrub in step 2) plays a minor role on the final GPP flux at 0.05° (see lower panels in **Supplementary Fig. S5**).

It is worth emphasizing that we assumed vegetation exists over urban areas and calculated relative fractions in steps 1) and 2) as the information of vegetated/impervious fractions is embedded in the 0.05° CSIF product (**Appendix A**). Additional information about vegetated/impervious fractions was not necessary in the calculation of $\alpha$ for every 500 m urban grid. For





example, if half of the 0.05° urban grid cell is covered by impervious land with the rest half covered by DBFs, SIF of this grid cell would already be lower than SIF of a grid cell fully covered by DBFs. Thus, instead of calculating a weighted mean $\alpha$ using slopes and land fractions of DBF and impervious land (whose $\alpha = 0$), we only need to assign the $\alpha$ of DBF to this particular 500 m urban grid. Otherwise, the resultant urban GPP (= CSIF * $\alpha$) could be underestimated by "double-counting" 5 of impervious fraction in both SIF and $\alpha$.

### 2.3 R$_{eco}$ estimates

Three explanatory variables—$T_{air}$, $T_{soil}$, and GPP— are chosen to train against the observed daily mean R$_{eco}$ from FLUXNET. During the training process, we built separate sets of NN models using 1) pure temperatures and GPP observations from EC towers, 2) ERA5 reanalysis and SIF-based GPP, or 3) Daymet with NLDAS and SIF-based GPP to account for mismatches in 10 reported soil depths (introduced in **Sect. 2.1.4**). SIF-based GPP is ingested in the hope to pass SIF information onto R$_{eco}$ estimation.

For each set of NN model, we manually split data based on their biomes and obtain 12 separate NN models. Data points from open and closed shrublands are combined due to the extremely low numbers of EC sites around the globe. To be consistent 15 with the C3/C4 crops partition for GPP estimates (**Sect. 2.2.1**), we obtained 2 separate NN models for C3 and C4 crops and calculated the weighted mean R$_{eco}$ based on the derived C3:C4 ratios. For each biome type, 80 % and 20 % of the data points are used for training and testing, respectively. Models with two hidden layers are constructed with 32 and 16 neurons chosen for the first and the second layer, respectively. We computed R$_{eco}$ at 500 m by applying biome-specific models and then calculated the biome-weighted mean R$_{eco}$ at 0.05°. To some extent, this process facilitates the R$_{eco}$ estimates over urban areas 20 (that comprise various possible vegetation types) and croplands and reduces possible biases in final 0.05° R$_{eco}$. We also tested two other ways to train R$_{eco}$ based on 1) all data points without differentiating their land cover types and 2) additional categorical variables from biomes and month and season of the year. Please refer to **Appendix B** for sensitivity tests and technical details about data preparation and cross validation of neural networks. Comparison of modeled R$_{eco}$ against FLUXNET is shown in **Sect. 3.2**.

25 ### 2.4 NEE estimates with hourly and spatial downscaling

We obtained hourly surface downward shortwave radiation (SW$_{rad}$) and air temperature ($T_{air}$) from the latest ERA5 reanalysis for calculating the hourly scaling factors for GPP and R$_{eco}$. Both $T_{air}$ and SW$_{rad}$ are provided on the horizontal grid spacing of 0.25° and then bilinearly interpolated to the finer grid spacing. To estimate the hourly radiation scaling factors (i.e., $I_{scale}$) for GPP, we normalized the hourly SW$_{rad}$ by the 4-day mean SW$_{rad}$ that matches the 4-day mean GPP. Regarding the calculation 30 of hourly temperature scaling factors (i.e., $T_{scale}$) for R$_{eco}$, a temperature sensitivity function ($SR_{1h}$) has been modified from prior studies (Fisher et al., 2016; Olsen and Randerson, 2004):





$$SR_{1h} = Q_{10}^{\frac{T_{air,1h}-30°C}{10°C}}, \tag{1}$$

where $Q_{10}$ is an unitless temperature sensitivity parameter that could vary across biomes, and $T_{air}$ is in °C. Because the hourly downscaling procedure is performed on GPP and $R_{eco}$ fluxes at 0.05° and no single biome is associated with the 0.05° grid cell, we simply adopt a typical $Q_{10}$ value of 1.5 according to previous studies (Fisher et al., 2016) despite potential variations in

$Q_{10}$. After normalizing $SR_{1h}$ by its daily mean value to match the modeled daily mean $R_{eco}$, we use $I_{scale}$ and $T_{scale}$ to downscale $R_{eco}$ and GPP fluxes, i.e., $\overline{R_{eco,1day}}$ and $\overline{GPP_{4days}}$:

$$R_{eco,1h} = \overline{R_{eco,1day}} * T_{scale} = \overline{R_{eco,1day}} * \frac{SR_{1h}}{\frac{1}{24}\Sigma_{1day} SR_{1h}},$$

$$GPP_{1h} = \overline{GPP_{4days}} * I_{scale} = \overline{GPP_{4days}} * \frac{SW_{rad,1h}}{\frac{1}{24*4}\Sigma_{4days} SW_{rad,1h}}. \tag{2}$$

Examples of $I_{scale}$ and $T_{scale}$ on July 2, 2018 over the western US is displayed in **Supplementary Fig. S6bc**. $I_{scale}$ is simply zero

between 04:00 and 14:00 UTC for the western US while $T_{scale}$ is usually smaller during nighttime than daytime. Consequently, plants begin to photosynthesize starting at around 14:00 UTC (07:00 – 08:00 local time on this particular day), while net biogenic fluxes may remain positive due to opposing effects from $R_{eco}$. As a sanity check of the ERA-based $SW_{rad}$, we examined an alternative product at a higher grid spacing than ERA5, of which $SW_{rad}$ and PAR were estimated based on the Earth Polychromatic Imaging Camera (EPIC) data onboard the Deep Space Climate Observatory (DSCOVR) and random

forest algorithms (dx.doi.org/10.25584/1595069). This EPIC-based SW and PAR model estimates are available globally at 10 km from June 2015 to present and has been validated against in situ observations from the Baseline Surface Radiation Network (BSRN) and Surface Radiation Budget Network (SURFRAD). Hourly $I_{scale}$ calculated using ERA5-based $SW_{rad}$ and EPIC-based $SW_{rad}$ or PAR zoomed into different cities generally agree well in terms of their mean diurnal variations with small discrepancies in the maximum values during noon hours (**Supplementary Fig. S6d**) but play a minor role on resultant GPP

diurnal cycles (**Supplementary Fig. S6e**).

As an optional step in SMUrF, one can choose to generate NEE fluxes at 1/120° with examples shown in **Supplementary Fig. S7**. We followed the bilinear interpolation method (Ye et al., 2020) but used MODIS VCF to provide spatial reallocation matrix. We decided to stop at 1/120° and not spatially redistribute NEE down to 250 m due to increasing uncertainties.

**2.5 Uncertainty quantification and direct flux validation**

Error quantification is important for characterizing the precision and accuracy of modeled fluxes. Previous studies have carried out comprehensive uncertainty estimates towards their reported biogenic flux estimates (Dietze et al., 2011; Hilton et al., 2014; Lin et al., 2011; Xiao et al., 2014). Here the uncertainty in SMUrF-based NEE is comprised of random uncertainties from GPP and $R_{eco}$. Even though the $R_{eco}$ calculation relies on the use of GPP, we assume no correlation for simplification and sum up

the error variances from GPP and $R_{eco}$ as the NEE uncertainty at the daily scale.





We first extrapolated any dependent variables, including the MODIS-based biome type, 4-day mean SIF, and daily mean $T_{air}$ & $T_{soil}$ from their gridded fields onto each flux tower location and computed directly the modeled GPP and $R_{eco}$ using $\alpha$ value and te fitted NN models. A comparison between these direct computations and screened observations from FLUXNET2015 is carried out to yield the biome-specific root mean square error (RMSE), mean bias, and coefficient of variation (CV).

Eventually, biome-specific CVs are related to each 500 m grid cell and aggregated as weighted mean CV at 0.05° in a root-mean-square error manner assuming statistical independence. Therefore, an uncertainty is associated with each 0.05° modeled flux at the daily scale. Despite that GPP and $R_{eco}$ are predicted based on biomes, we collected model-data pairs from all biomes and displayed them as density plots in **Fig. 5**, for purposes of presentation.

Directly computed 4-day mean GPP at most towers match well with observations regarding their magnitude and seasonality. Modeled GPP shows a slight underestimation against irrigated maize sites in Nebraska (e.g., US-Ne1, US-Ne2) and sites in the central valley (e.g., US-Twt) in California (**Supplementary Fig. S8**) because irrigation effect is not implicitly included and high crop chlorophyll concentration may not be fully recreated in reflectance-driven CSIF data (**Sect. 2.1.2**). Nevertheless, the overall correlation coefficient between directly computed modeled GPP from SMUrF and partitioned mean GPP from

FLUXNET is ~0.86 with a mean bias of –0.069 $\mu$mol m$^{-2}$ s$^{-1}$ for all 89 tower sites. When removing cropland sites from consideration, RMSE in 4-day mean GPP drops from 1.91 to 1.74 $\mu$mol m$^{-2}$ s$^{-1}$ (**Fig. 5a** vs. **5b**).

We also reported prediction performances of $R_{eco}$ over testing sets (i.e., 20 % of the entire data volume). $R_{eco}$ trained and predicted using input variables from FLUXNET overperform the ones using ERA5's temperatures and SIF-modeled GPP (r =

0.90 vs. r = 0.87; **Fig. 5c vs. 5d**). The constant SIF within a 4-day interval and the constant $\alpha$ over seasons make it more difficult to reproduce daily variations in $R_{eco}$ as only $T_{air}$ and $T_{soil}$ are varying. Recall that temperature fields from higher resolution fields (Daymet + NLDAS) were also used for training and predicting $R_{eco}$ over CONUS. These Daymet + NLDAS runs (**Fig. 5g**) appear to slightly outperform ERA5 runs (**Fig. 5f**) but underperform compared to runs using FLUXNET variables (**Fig. 5e**), according to error statistics like r, RMSE, and mean bias. Although the NN model using FLUXNET-based

GPP and temperatures returns a better performance, we are inclined to use NN models trained by modeled features for two reasons: 1) to account for discrepancies in GPP and temperature between tower observations and model/reanalyses and 2) for spatial generalization beyond points with "ground truth" as only modeled GPP are available away from the EC sites. Nevertheless, the mean bias on the testing set for all biomes remains small (< 10$^{-3}$ $\mu$mol m$^{-2}$ s$^{-1}$) with the overall RMSE of ~1.14 $\mu$mol m$^{-2}$ s$^{-1}$. Since the error statistics based on NN models driven by ERA5 resemble the ones based on NN models

driven by Daymet/NLDAS (**Figs. 5f vs. 5g**), we will only present ERA5-based outputs throughout the result section (**Sect. 3**).

We stress that directly computed fluxes and their validation against FLUXNET presented in this section differ from those presented in **Sect. 3.2.1**, as the latter one is the spatially weighted mean flux at 0.05° that takes the spatial heterogeneity into account.





## 2.6 Comparisons of FFCO₂ vs. NEE fluxes and their contributions in column CO₂

FFCO₂ emissions from the Open-Data Inventory for Anthropogenic Carbon dioxide (ODIAC2019, Oda et al. 2018) are compared against NEE from SMUrF in terms of their seasonal magnitudes, summertime diurnal cycles, and spatial distribution (result in **Sect. 3.1**). Initial ODIAC emissions at 1 km grid spacing have been averaged to 0.05° to match SMUrF for such

FFCO₂-NEE fluxes comparisons. Despite that the city-wide FFCO₂ emissions from ODIAC may differ from other reported emissions (Chen et al., 2020; Oda et al., 2019), ODIAC has been widely adopted and provides sufficient insights to FFCO₂ emissions from a global perspective.

To further translate CO₂ fluxes into changes in column-averaged CO₂ concentrations, or XCO₂ (**Sect. 3.3**), we made use of an

atmospheric transport model—i.e., the column version of the Stochastic Time-Inverted Lagrangian Transport (X-STILT, Wu et al., 2018) model. We carried out four case studies by examining summertime OCO-2 tracks over Boston, Indianapolis, Salt Lake City, and Rome. Those 4 cities are chosen based on data availability and quality and their different vegetation coverages. Specifically, thousands of air parcels are released in STILT (Fasoli et al., 2018; Lin et al., 2003) from the same atmospheric columns as the OCO-2 soundings and driven by meteorological fields, i.e., the 3-km High Resolution Rapid Refresh (HRRR)

for US cities and 0.5° GDAS for non-US cities (Rolph et al. 2017). The X-STILT model returns hourly surface influence matrices or "column footprints" that reveal the influence of each upwind grid cell onto downwind satellite soundings within each hour interval. These hourly footprints can then be convolved with surface fluxes, e.g., hourly SMUrF-based NEE or ODIAC-based FFCO₂ emissions, to reveal spatially-explicit XCO₂ contributions due to biogenic or anthropogenic fluxes along with total XCO₂ anomalies (XCO₂.bio, XCO₂.ff) at each receptor if summing up aforementioned spatial contributions. For more

details of ODIAC and X-STILT, please refer to Oda et al. (2018) and Wu et al. (2018), respectively.

## 3 Results

We start with model outputs of biogenic and anthropogenic fluxes at the regional and urban scales (**Sect. 3.1**) and their comparisons against EC observations and other models over natural biomes and urban areas (**Sect. 3.2**). Although the SMUrF model can be used to estimate NEE fluxes globally, for the purposes of this paper gridded fluxes at 0.05° were produced from

Jan 1, 2020 to Dec 31, 2018 only for the following populated and vegetated regions: CONUS, western Europe, east Asia, south America, central Africa, and eastern Australia.

## 3.1 Biogenic and anthropogenic CO₂ fluxes at the regional and city scale

To reveal the relative importance of biospheric fluxes in the context of anthropogenic emissions, we summed up 3-month mean SMUrF-based NEE fluxes and ODIAC-based FFCO₂ emissions over CONUS, western Europe, and east Asia (**Fig.**

**6abc).** If we first focus on the areas that are shaded in green in regional scale panels indicating net carbon uptake, SMUrF is able to reveal the spatial contrasts and seasonal variations in its NEE fluxes, as informed by the use of SIF, land cover types,



and temperature fields. Places with strong seasonal amplitude are found to be rural regions covered by crops and dense forests, e.g., eastern US, northeastern and southern China, and spotty locations over Europe (**Fig. 6**). Even after adding $FFCO_2$ emissions, "brownish" spots appear all over the map, in particular over eastern China where the sum of NEE and $FFCO_2$ stay positive even in summer months (**Fig. 6c**).

We next zoom into cities. SMUrF captures the increasing biospheric activities from urban cores to their rural surroundings inferred by GEE, $R_{eco}$, and NEE components (eight zoomed-in panels in **Fig. 6**), as cities are usually associated with less vegetation coverage than their rural counterparts. The urban-rural difference in GEE over Salt Lake City, Boston, and Seoul is relatively small, in contrast to cities like Guangzhou and Tokyo. Since modeled $R_{eco}$ is partially driven by SIF-based GPP,

the spatial variations of GEE and $R_{eco}$ appear alike to some extent. Even though average GPP can be high over JJA 2018, the summertime mean NEE in the urban core remains small, with values ranging from $-1$ to $-2$ $\mu$mol m$^{-2}$ s$^{-1}$. Spatial distributions of GEE and $R_{eco}$ for Boston derived from SMUrF are similar to what have been reported in Hardiman et al. (2017) using urbanVPRM; both models predict relatively strong biospheric uptake even within the Boston urban core (top right panel in **Fig. 6**). $R_{eco}$ with the same sign as $FFCO_2$ emission is more spatially uniform and exceed $FFCO_2$ over residential and rural

areas away from the urban cores, which is consistent with Decina et al. (2016), who found that the soil respiration component of $R_{eco}$ can be comparable to $FFCO_2$ over the residential and forest regions (i.e., > 10 km) to the west of Boston center. Yet, when it comes to interpreting observed $CO_2$ concentrations, it is the net flux (i.e., NEE, not $R_{eco}$) that should better be evaluated against $FFCO_2$. In short, within these eight urban cores $FFCO_2$ rather than NEE control the spatiotemporal patterns of carbon exchange. However, over residential or rural areas, biogenic $CO_2$ fluxes have the potential to dominate carbon exchange.

We further extend the analysis to 40 cities around the globe to help reveal how carbon fluxes vary between 1) urban vs. adjacent rural areas, 2) different cities, 3) the FF vs. NEE components and 4) over seasons. Specifically, we examine spatially average biogenic and anthropogenic $CO_2$ fluxes over "urban" and "rural" grid cells within a 2° × 2° region around city centers in terms of seasonal variation during 2017 – 2018 (**Sect. 3.1.1**) and mean diurnal cycle over JJA 2018 (**Sect. 3.1.2**). Urban grids are

defined as the "urban and build-up settlements" according to MCD12Q1, while rural grids contain all the natural counterparts (e.g., forests, grasslands, croplands) except for water, ice, and barren lands. The diurnal cycles of $FFCO_2$ are calculated using the hourly scaling factors from Temporal Improvements for Modeling Emissions by Scaling (TIMES, Nassar et al., 2013) on top of monthly mean ODIAC-derived emissions. Note the TIME emission temporal patterns are climatological, based on a U.S gridded inventory by Gurney et al (2009), and thus do not change in response to local environmental conditions, such as

air temperature.

### 3.1.1 Seasonal variation

More negative NEE during growing seasons and stronger seasonal amplitude are mainly linked to rural grids than to urban grids as expected (**Fig. 7a** vs. **7b**). Among our 40 selected cities, the top "wet", biologically active cities with minimum 4-day





mean NEE below –2 $\mu$mol m$^{-2}$ s$^{-1}$ include Boston, Baltimore-DC, New York, Taipei, London, Paris, and Rio de Janeiro. By contrast, a few "dry" cities stand out, with their average NEE over urban grid cells close to or slightly above zero, such as Los Angeles, Phoenix, and Madrid (**Fig. 7a**). Besides magnitude, cities achieve their maximum net sink at different times in the year— e.g., June/July for most cities in eastern US and east Asia (except for Taipei); a slight earlier peak for most cities in
western US, western Europe, and Taipei; and January for cities in Australia, South America, and Kinshasa in central Africa. For instance, the maximum biospheric sink in London is found in late May to June for both 2017 and 2018 (**Fig. 7a5**), which is consistent with the seasonality of the posterior NEE fluxes reported for UK from 2013 to 2014 (White et al., 2019). SMUrF may pick up interannual variations in its estimated fluxes primarily owing to changes in model drivers (**Fig. 7**).

Since FFCO$_2$ emission varies less over seasons than NEE, we focus comparing the magnitude of the seasonal amplitude in NEE relative to the annual mean FFCO$_2$, shown as numbers printed below the city names in **Fig. 7**. Such comparison helps to inform the potential "interference" from the biosphere within and around cities towards interpreting long-term CO$_2$ observations, albeit without considering the atmospheric transport. The comparison reveals that FFCO$_2$ over most urban grids are stronger than the seasonal amplitude in NEE (**Fig. 7a**). Exceptions include Pyongyang and cities in central Africa whose
seasonal NEE amplitudes approach their annual mean FFCO$_2$ emissions, e.g., ~2.0 vs. 2.5 $\mu$mol m$^{-2}$ s$^{-1}$ for Lagos. Things can get more complex if one tries to interpret FFCO$_2$ signals from year-long observations over rural areas given the fact that their annual mean FFCO$_2$ emissions seldom exceed 2 $\mu$mol m$^{-2}$ s$^{-1}$, whereas seasonal amplitudes in their 4-day mean NEE are often > 5 $\mu$mol m$^{-2}$ s$^{-1}$, except for Chicago, Taipei, PRD, Shanghai and Busan.

### 3.1.2 Diurnal cycle

In regard to the timing of hourly fluxes, NEE in most mid-latitude cities start to dip below 0 at ~7 or 8 am local time (**Fig. 8a**), slightly later than the typical summer sunrise hour of ~6 am, with a lag associated with the time it takes for GEE to offset R$_{eco}$. Meanwhile, FFCO$_2$ emissions start to rise due to morning traffic. NEE reaches its minimum at different hours between cities, spanning from 11 am to 1 pm before air temperature reaches its maximum. NEE rises back to zero before 6 pm, earlier than the typical summer sunset time of ~8–9 pm, considering the time lag between air temperature drop and shut-down of solar
radiations. In Boston, SMUrF NEE sees a few hours delay compared to urbanVPRM (Hardiman et al., 2017), which reported NEE turning negative at ~5 am local time in July 2013.

With regard to the magnitude of hourly fluxes, FFCO$_2$ emissions is negligible for rural grids compared to cases of urban grids (**Fig. 8b**). Within the east Asian urban cores, intensive FFCO$_2$ emissions ranging from 20 to 60 $\mu$mol m$^{-2}$ s$^{-1}$ play a dominant
role in the total CO$_2$ fluxes even during noon hours (**Fig. 8a3-4**). As for a few cities in eastern US and Europe, the urban biosphere may take up a considerable amount of CO$_2$, which approaches or even exceeds FFCO$_2$ emissions during noon hours in summer months. For example, the mean maximum biospheric uptake flux and FFCO$_2$ emission over Boston is about –16 $\mu$mol m$^{-2}$ s$^{-1}$ and 10 $\mu$mol m$^{-2}$ s$^{-1}$, respectively. Compared to the diurnal NEE and FFCO$_2$ estimated for Boston by urbanVPRM




(Hardiman et al., 2017), our reported mean FFCO$_2$ is much weaker with stronger biospheric uptake, which result from differences in spatial extents for averaging fluxes between two studies. For instance, Hardiman et al. (2017) chose a much smaller area (~25 km x 25 km) around Boston that yields much higher FFCO$_2$ emissions (up to 30 kg-C ha$^{-1}$ hr$^{-1}$).

## 3.2 Flux comparisons

The aforementioned analyses have revealed the urban-rural gradient in both FF and NEE fluxes regarding their magnitude and timing at various spatiotemporal scales. As a step forward, we will see how robust our modeled fluxes are at the hourly scale and how the FF and NEE fluxes impact the CO$_2$ concentrations downstream after considering atmospheric transport (**Sect. 3.3**). In the following subsections, simulated hourly mean NEE fluxes are extracted directly from gridded output fields compared against non-gapfilled observed NEE at dozens of EC sites from FLUXNET and INFLUX, which differ from the

comparison based on directly computed daily mean fluxes in **Sect. 2.5**.

### 3.2.1 SMUrF vs. FLUXNET and INFLUX

We first look at how SMUrF prediction perform as compared against 67 EC tower sites from FLUXNET2015 in North America and Europe from 2010 to 2014 (**Fig. 9**). We are aware of the potential mismatch as performing flux comparisons between grid-level model output and point-level observations. As land cover types around EC sites can be heterogenous within the 0.05°

model grid, we computed the areal fractions of the specific land cover type indicated by EC sites over the total grid area inferred from MCD12Q1. One can refer to the areal fractions for individual sites in **Supplementary Fig. S9.**

The correlation coefficient between simulated and measured hourly NEE for most biome types ranges from 0.66 to 0.79, except for open shrubland (r = 0.43), likely due to limited amount of data. Higher random and systematic uncertainties are associated

with these hourly flux comparisons than direct daily validations shown in **Sect. 2.5** (**Fig. 5**), due to larger flux magnitude and increasing uncertainties arisen from hourly downscaling and possible heterogeneity of land cover type in ~5 km model grid. The mean bias of hourly NEE ranges from -1.51 $\mu$mol m$^{-2}$ s$^{-1}$ for grassland to +1.11 $\mu$mol m$^{-2}$ s$^{-1}$ for closed shrubland, with only one site available globally. Among all biomes, random uncertainties as indicated by the RMSE are the highest over croplands, which stem from their inter-site variations in GPP-CSIF relationship (**Supplementary Fig. S1**) and strong fluxes.

The potential underestimation in 4-day mean GPP over irrigated maize sites (e.g., US-Ne* as mentioned in **Sect. 2.5**) seems to get propagated into the NEE estimate, as modeled hourly biospheric uptakes during daytime are weaker than measured uptakes (**Supplementary Fig. S10a**). Nevertheless, if treating all crop sites together, the timing and magnitude of the 3-month mean diurnal cycle of NEE from SMUrF resembles those from observations (1$^{st}$ column of **Supplementary Fig. S10b**).

Besides FLUXNET2015, we carried out independent comparisons against EC sites from INFLUX network in and around the city of Indianapolis (**Fig. 10a, Sect. 2.1.5**), whose observed fluxes were not used for calibrating SMUrF's parameters. Site #3, covered by corn, has a stronger observed uptake from mid-June to mid-July than site #2 with mixed crops (**Figs. 10d**), because





corn is often associated with higher light saturation points and lower light compensation points, leading to higher light use efficiency and GPP than soybean. Although C3-C4 fractional contribution was incorporated into the calculation of $\alpha$ (**Sect. 2.2.1**), SMUrF is unable to differentiate NEE at two adjacent crop sites found essentially in the same 0.05° model grid, simply because SIF is only available at 0.05°. The simulated NEE may agree better with the average observed NEE of two crop sites.

Hourly measured NEE at crop sites range from –64.4 to 28.1 $\mu$mol m$^{-2}$ s$^{-1}$ while simulated values span from –66.7 to 12.1 $\mu$mol m$^{-2}$ s$^{-1}$ with mean bias of –0.63 $\mu$mol m$^{-2}$ s$^{-1}$ and RMSE of 6.6 $\mu$mol m$^{-2}$ s$^{-1}$ over the entire observed window (**Fig. 10d**). Absolute mean bias and RMSE derived from the two independent crop EC sites outside Indianapolis are in proximity to or slightly higher than those derived from FLUXNET crop sites—i.e., bias of –0.63 vs. –0.39 $\mu$mol m$^{-2}$ s$^{-1}$ (**Fig. 10c** vs. **Fig. 9**).

Lastly, we focus on independent comparisons at EC sites deployed within the city of Indianapolis. The first thing to notice is that NEE fluxes at sites #1 and #4 exhibit a seasonally attenuated pattern but slightly stronger biospheric activities from Nov and Dec compared to the crop sites (**Fig. 10de**). The correlation coefficient of hourly NEE fluxes is ~0.75. Modeled mean diurnal cycles over JJA 2018 coincide with those from observations at urban vegetation sites (**Fig. 10f**).

### 3.2.2 SMUrF vs. UrbanVPRM for Los Angeles

The comparison of SMUrF to dozens of EC sites have yet to offer much insights on the spatial distribution of urban $CO_2$ fluxes within an urban region. Therefore, we further compare SMUrF against the spatial distribution of fluxes derived from an especially high-resolution urbanVPRM simulation (**Sect. 2.1.5**) at 30-m grid spacing over Los Angeles, from July to Sept 2017. SMUrF and urbanVPRM agree in regards to the spatial distribution of estimated GEE fluxes (**Fig. 11a**). For instance, both simulate stronger uptakes in more vegetated mountainous and residential areas to the northeast of Pasadena and San Bernardino,

with nearly zero uptake over the Moreno Valley to the south of Riverside and the LA basin. SMUrF models stronger biospheric uptake than urbanVPRM across LA over JAS in 2017 (1$^{st}$ column in **Fig. 11a**).

Both models have incorporated observed GPP from EC sites for tuning their parameters, but the two models are driven with different spatial proxies (SIF vs. EVI and LSWI) and adopt different model formulations. As one of the key improvements,

urbanVPRM revises the initial VPRM-based $R_{eco}$ by incorporating impervious surface areas (ISA), which in turns modify the air temperature over urban cores due to urban heat island ($T_{UHI}$) effect and affect estimated GPP, autotrophic and heterotrophic respirations. For example, $R_H$ can be reduced while $R_A$ and GPP may be increased given enhanced $T_{UHI}$ over higher ISA regions in urbanVPRM. Regardless, $T_{UHI}$-revised $R_{eco}$ in urbanVPRM still follows a simple function of air temperature. ISA is implicitly contained in SIF although not explicitly considered in SMUrF. $R_{eco}$ in SMUrF is driven not only by air temperatures but also soil temperatures and GPP. Thus, $R_{eco}$ in SMUrF appears to be spatially more correlated with its GPP. These methodological discrepancies lead to an overall higher $R_{eco}$ and negative NEE over LA for SMUrF than urbanVPRM (3$^{rd}$ column in **Fig. 11a**). Similarly, the 3-month mean diurnal cycles of hourly NEE extracted from a few grids indicate stronger daily amplitudes according to SMUrF (**Fig. 11b**). Both models indicate Pasadena, towards the northern end of LA, is associated





with a stronger diurnal amplitude than downtown LA, with discrepancies in NEE during noon hours likely due to differences in the data products from which PAR or $SW_{rad}$ are derived (e.g., due to cloud coverage and spatial resolution).

Despite the opposing signs between urbanVPRM and SMUrF modeled NEE over downtown LA, the overall biological
activity (either net positive or negative) remains small, particularly when compared against $FFCO_2$ emissions. As a quick analysis, we defined "downtown LA" as a simple rectangle with the following lat/long boundaries: 118.5°W–118°W and 33.9°N – 34.1°N and removed one gridcell with intensive $FFCO_2$ emissions from consideration (likely due to point sources, **Fig. 11c**). The 3-month mean $FFCO_2$ around downtown LA is ~12.1 $\mu$mol m$^{-2}$ s$^{-1}$, while the relative differences in NEE between two biogenic models remain small, at ~0.41 $\mu$mol m$^{-2}$ s$^{-1}$.

**3.3 Urban-rural gradient in $XCO_{2.ff}$ and $XCO_{2.bio}$**

After evaluating hourly NEE and informing their urban-rural contrast around 40 cities, we examine the imprint of urban-rural NEE contrasts on $CO_2$ concentrations. As described in **Sect. 2.6**, we analyzed OCO-2 $XCO_2$ observations over a few cities and accounted for the atmospheric transport between upwind carbon sources/sinks and downwind satellite soundings as reflected by X-STILT's column footprints (**Fig. 12a**). Only places with non-zero footprints and non-zero fluxes contribute to anomalies
in downwind $XCO_2$, which can be visualized by the spatial $XCO_{2.ff}$ enhancements and $XCO_{2.bio}$ anomalies in **Figures 12c** and **12d**. Eventually, the sum of those spatial $XCO_2$ anomalies serves as the total anthropogenic or biogenic $XCO_2$ anomalies per receptor (orange or green dots in **Fig. 12b**).

Boston, surrounded by forests, provides the main case study here as its anthropogenic and biogenic fluxes have been
extensively studied by previous studies (Decina et al., 2016; Hardiman et al., 2017; Sargent et al., 2018), notwithstanding the small number of qualified OCO-2 tracks. On July 7, 2018, the northeasterly wind transported air from Boston to downwind satellite soundings (**Fig. 12a**). Most biogenic contributions stayed negative due to daytime photosynthetic uptake (overpass time of ~1300 LT), especially over the area immediately around soundings (dark green colors in **Fig. 12d**) where strong footprints are found (green-yellow colors in **Fig. 12a**). During the few hours prior to 1300 LT, air parcels within the planetary
boundary layer began to encounter morning-time positive NEE away from the receptors, which led to slightly positive $XCO_{2.bio}$ contributions (light yellow in **Fig. 12d**). Still, strong negative near-field anomalies prevail over the slightly positive far-field anomalies, which result in the overall negative $XCO_{2.bio}$ per receptor spanning from –2 ppm to –0.3 ppm (green dots in **Fig. 12b**). The convolution between footprints and $FFCO_2$ emissions is simpler, always yielding overall positive enhancements (orange colors in **Fig. 12c**), especially over soundings from 41.7 ° to 42.3° N with overall $XCO_{2.ff}$ enhancements of up to 0.6
ppm (orange dots in **Fig. 12b**).

We then take a closer look at $XCO_2$ anomalies within three latitude bands (colored ribbons in **Fig. 12b**)—i.e., 1) ocean (41°– 41.5° N), 2) rural (41.5°– 41.7° N), and 3) urban-enhanced region (41.7 °– 42.1° N). Simulated $XCO_2$ anomalies over the ocean





soundings are close to zero due to minimal upwind influences from FFCO$_2$ or NEE. Moving northward, soundings within the rural band experience intensive biogenic activities that lead to total signal of up to –2 ppm. Measurements within the urban-enhanced region differ from the previous two. Due to a less active urban biosphere than its surrounding forests, a rise in XCO$_{2.bio}$ has been spotted centered at ~41.9° N (**Fig. 12b**) relative to the adjacent rural band. That being said, the urban-rural

gradient in XCO$_{2.bio}$ anomalies (hereinafter "ΔXCO$_{2.bio}$") can exceed 0.5 ppm, which is comparable to the XCO$_{2.ff}$ of ~0.6 ppm. We will more thoroughly discuss the implication of this urban-rural bio-gradient ΔXCO$_{2.bio}$ in the context of the total measured XCO$_2$ in **Sect. 4.1** (**Fig. 12e**).

Clearly, ΔXCO$_{2.bio}$ can vary with locations (upwind vs. downwind geometries) and time of day or year when measurements

were taken. Thus, we attempted to study a few more OCO-2 tracks near Indianapolis, Salt Lake City, and Rome given their different surrounding vegetation types (**Fig. 13**). These summertime tracks were picked based on their richness in screened soundings (QF = 0) that facilitates the X-STILT modeling. Although the OCO-2 track is adjacent to Indianapolis, upwind regions relative to downwind soundings are located over the east and away from Indianapolis leading to minimal XCO$_{2.ff}$ anomalies while strong biospheric uptake and XCO$_{2.bio}$ below –2 ppm (1[st] row in **Fig. 13**). Strong influence with the surface

land is concentrated over the Salt Lake Valley on June 25, 2018, leading to a maximum XCO$_{2.ff}$ of about 0.5 ppm and XCO$_{2.bio}$ with comparable magnitude but negative signs (2[nd] row in **Fig. 13**). For Rome, strong footprints to the west of the city center interact with positive NEE, which give rise to a maximum XCO$_{2.bio}$ of up to 1.5 ppm (3[rd] row in **Fig. 13**) over 42°– 42.5°N where XCO$_{2.ff}$ can be significant as well. To sum up, XCO$_{2.bio}$ anomalies can be either positive or negative with an absolute magnitude reaching about 2.5 ppm during the growing season, as seen in the cases of Indianapolis and Boston. The urban-

rural bio-gradient ΔXCO$_{2.bio}$ is smaller, with maximum values ranging from 0.6 to 1.0 ppm among our limited cases, as implied by green dots in the 2[nd] column in **Figure 13**. Based on our limited cases, XCO$_{2.bio}$ anomalies may be less negative (more positive) within the urban-enhanced region, serving as a peak that coincides with the XCO$_{2.ff}$ peak. More cities and satellite tracks across the globe should be studied to generalize the conclusion.

## 4 Discussion and summary

Finally, we discuss applications and limitations of SMUrF. In particular, we examine how urban-rural bio-gradient in XCO$_{2.bio}$ may alter the interpretation of XCO$_2$ observations during the growing seasons.

### 4.1 Implications on background XCO$_2$ determination

As the extraction of urban emissions from column measurements can be quite sensitive to the background definition (Wu et al., 2018), we illustrate the impact of urban-rural gradient in XCO$_{2.bio}$ during the growing season in the context of

background determinations. First, let us consider the background XCO$_2$ (XCO$_{2.bg}$) defined as the average of observations over the region outside the city unaffected by FFCO$_2$. Observed enhancements are then calculated as levels of XCO$_2$





elevated above $XCO_{2.bg}$. This mean constant $XCO_{2.bg}$ reasonably represents the $XCO_2$ portion unaffected by urban emissions during non-growing seasons for most places (Wu et al., 2018). However, this definition of $XCO_{2.bg}$ might implicitly neglect the variation in biospheric $XCO_2$ anomalies between the urban-enhanced versus the background region. Again, it is the gradient in $XCO_{2.bio}$, not its absolute level, along satellite tracks that modify the background value.

The goal, then, becomes to recalculate a latitude-dependent background—$XCO_{2.bg}$ (lat) in **Eq. 3**—that integrates adjustment of the biospheric gradient, referred here as $\Delta XCO_{2.bio}$. To quantify the $\Delta XCO_{2.bio}$ correction term, we average $XCO_{2.bio}$ within the background latitude band ($XCO_{2.bio.bg}$) and subtract it from the latitude-varying $XCO_{2.bio}$. The correction term is then added to the constant background ($XCO_{2.bg.const}$) to yield a latitudinally varying, bio-adjusted background:

$$XCO_{2.bg} \text{ (lat)} = XCO_{2.bg.const} + \Delta XCO_{2.bio} \text{ (lat)}$$
$$= XCO_{2.bg.const} + XCO_{2.bio} \text{ (lat)} - XCO_{2.bio.bg}. \tag{3}$$

Recall that $XCO_{2.bio}$ (lat) is the modeled biospheric anomalies using SMUrF and X-STILT (**Sects. 2.6** and **3.3**).

To facilitate visualization and understanding of $\Delta XCO_{2.bio}$ and bio-adjusted background, let us return to the Boston case again (**Fig. 12**). Here we only focus on the soundings north of 41.5°N to avoid the possible upwind influence from oceanic $CO_2$ flux and its gradient. We treat the rural band (41.5°– 41.7° N, light green ribbon in **Fig. 12e**) as the background region, given minimal influences from urban emissions. The constant background is 403.60 ppm (dark green line in **Fig. 12e**), with mean $XCO_{2.ff}$ and $XCO_{2.bio}$ anomalies of about 0.19 ppm and –1.33 ppm, respectively, within the background region. After integrating the bio-gradient $\Delta XCO_{2.bio}$, the new bio-adjusted background varies along latitude (light green line). If modeled $XCO_{2.ff}$ is added to the bio-adjusted background, the resultant total $XCO_2$ (purple line) better reproduces the latitudinal variations of the measured mean values (black triangles), with a slight latitude shift due to possible wind bias in the modeled atmospheric transport. Nevertheless, both the observed $XCO_2$ and modeled $XCO_2$ correcting for $\Delta XCO_{2.bio}$ exhibit a dip in $XCO_2$ within the rural band that is missing from the model result using the constant background (orange line). Thus, clearly in this case neglecting the latitudinal/spatial gradient in biogenic $XCO_2$ anomalies given gradients in NEE affects the extracted urban signal and the inferred $FFCO_2$ emissions.

**4.2 Summary, limitation and application**

This study aims at offering a model representation of biogenic $CO_2$ fluxes—SMUrF—to help improve the $CO_2$ flux partitioning over cities worldwide. SMUrF takes advantage of SIF to derive GPP from urban to rural areas and incorporates multiple predictor variables to estimate $R_{eco}$ using a neural network approach. Here we identify several model limitations and room for future improvements.

First, in our efforts to predict fractions of different vegetated types using a general solution that can be widely applied to cities around the world, we fitted a broad relationship between relative tree fractions and AGB. Admittedly, this relationship is a





simplification proposed based on high-resolution data over Los Angeles and can be improved in future versions of SMUrF as more urban tree fractions data with high accuracy and resolution become available. Data from new satellite sensors can be used in the future that provide spatial SIF over cities, land surface temperatures, biomass, and height and canopy structures of forests at an unprecedented spatiotemporal resolution (Stavros et al., 2017). Examples of such satellite sensors include OCO-3 (Eldering et al., 2019), the ECOsystem Spaceborne Thermal Radiometer Experiment ECOSTRESS (Fisher et al., 2020) and the Global Ecosystem Dynamics Investigation LIDAR (GEDI, Duncanson et al., 2020), all on-board the International Space Station. These cutting-edge and future data streams may improve the approximation of tree types and fraction over cities.

Second, GPP is currently estimated within SMUrF as a linear function of SIF, using a set of constant biome-specific linear slopes ($\alpha$) without considering temporal or inter-site variations. The adoption of SIF has dramatically benefited and simplified the GPP calculation, since no extra satellite indices or impervious fractions need to be plugged in. However, further improvements in SIF products with higher spatiotemporal resolution and investigation of GPP-SIF relationship and its ecological drivers will facilitate more accurate modeling of GPP and $R_{eco}$. For example, $\alpha$ calculated based on CSIF appear to be more uncertain over crops and wetlands, resulting in higher uncertainty to estimated GPP (**Supplementary Fig. S1**). We also attempted to incorporate a C3/C4 ratio to $\alpha$ and GPP, but the outcome is limited by the spatial resolution of adopted data products. Besides, given that most global SIF products are only available for every 4 days or even longer temporal windows, the predicted daily $R_{eco}$ within a 4-day interval is driven by day-to-day variations in air and soil temperatures. Improvements in the temporal resolution of gridded SIF product (e.g., TROPOMI-derived SIF, Turner et al. 2020) will likely improve the prediction of GPP that gets propagated into the prediction of $R_{eco}$. More challenging, the current estimates of GEE and $R_{eco}$ over cities still depend on relationships derived using observations over natural biomes. Urban trees are found to possess different characteristics from natural trees (e.g., Smith et al., 2019), which can pose a difficult task to model without more dedicated observations within urban environments.

Detailed parameters in SMUrF can be further fined tuned for more localized applications. For instance, while we are currently using global radiation fields at hourly scales from ERA5 given our global perspective, even higher resolution shortwave radiation or PAR data with higher accuracy in cloud cover estimates can be adopted instead. Further, the reference temperature used in this work (30° C) can be a bit higher than that (20° C) used in the calculation of maintenance respiration and (25°C) in the calculation of heterotrophic respiration in CLM. The $Q_{10}$ parameter (here = 1.5) also varies significantly in space and time. Moreover, a few secondary-order ecological and environmental variables may affect $R_{eco}$. For example, we tried adding soil moisture into the training system, but the improvement in model performance is not that evident.

We also hope to examine more cities and different times of the day in future studies to better study the relative biogenic and anthropogenic contributions to $XCO_2$ anomalies.



Lastly, we summarize several potential applications of SMUrF as follows.

1) *Improve the separation between biogenic and anthropogenic $CO_2$ fluxes for urban studies*. Regardless of the exact approach adopted for background definition, researchers can combine atmospheric transport models with fluxes from SMUrF to get estimates of the $CO_2$ anomalies due to biogenic flux exchanges, as shown in **Sect. 4.1**.

2) *Fill-in the urban gap and assist regional flux inversions*. The state-of-art terrestrial models that go into flux inversions usually include global models with relatively coarse resolution (e.g., 4° by 5°) with possible downscaling approaches. At the other end of the spectrum are highly localized models at fine resolution but sometimes requiring customization for individual cities. SIF-based fluxes from SMUrF may help bridge between the continental scale and the urban scale, with reasonable fine resolution and regional to global coverage.

3) *Understand how urbanization modifies the planet and environment.* An approach like SMUrF contributes to an understanding of how biogenic $CO_2$ fluxes vary among cities with different urban policies and emissions. SMUrF offers a quick solution to the biogenic fluxes within urban areas and their rural surroundings.





**Data availability**

SMUrF with the latest changes is stored as GitHub repository at https://github.com/wde0924/SMUrF. The exact version of the SMUrF model used in this work is archived on Zenodo (Wu et al., 2020b).

*Instructions for running SMUrF*: The user should start with main scripts of *main_script_\*.r* in sequence for computing gridded GPP, R$_{eco}$, and NEE, since the calculation of R$_{eco}$ using *main_script_Reco.r* relies on GPP generated using *main_script_GPP.r*. All model-related subroutines written in R are stored in SMUrF/r/src with model-required dependences in SMUrF/data. For most of the cases, the user does not need to modify these subroutines. Model developments are ongoing and please contact the corresponding author if one would like to obtain model outputs for other regions/cities.

*Required input data products for driving SMUrF* (data citation; parameter name in the model script).
1) Clear-sky SIF provided by CSIF (Zhang et al., 2018; *csif.cpath*);
2) AGB from GlobBiomass (Santoro et al. 2018; *agb.path*);
3) MCD12Q1 v006 (Friedl and Sulla-Menashe, 2019; *lc.path*), see README.md on GitHub repository for info on how
to download the exact format that SMUrF requires;
4) Hourly air temperature and shortwave radiation downwards from ERA5 reanalysis (Copernicus Climate Change Service Information, 2017; *TA\** and *SSRD\**).

*Other important data products used in this paper* (not prerequisites for driving SMUrF) include:
1) the high-resolution 60 m urban land cover classification product (Coleman, 2020) available from Mendeley Data;
2) the ODIAC emission data product available from the Global Environmental Database hosted by the Center for Global Environmental Research at the National Institute for Environmental Studies (Oda and Maksyutov, 2015);
3) the EC measurements from FLUXNET2015 (Pastorello et al., 2017).

**Appendix A. Comparison of CSIF with TROPOMI-based downscaled SIF and vegetation fraction from WUDAPT**

We carried out two tests to verify the accuracy and capability of CSIF. Firstly, CSIF is compared against a newly developed SIF product from TROPOMI over the Contiguous United States (CONUS) from June to August in 2018 (Turner et al. 2020 & personal communication with Alex Turner). This downscaled TROPOMI-based SIF product is initially available at 500 m and then averaged to 0.05° for the comparison. Due to the discrepancies in the reported SIF retrieval wavebands between OCO-2 (757 and 771 nm) and TROPOMI (740 nm), the OCO-2 based CSIF (757 nm) is scaled by an empirical scaling factor of 1.56
(Köhler et al., 2018) to yield comparison with TROPOMI SIF at the far-red SIF peak of 740 nm. Note that this scaling factor of 1.56 is only applied here for model comparisons and not for flux calculations. OCO-2 based CSIF and TROPOMI-based SIF both see high biological activities over the eastern CONUS and agree well spatially (**Supplementary Fig. S2**). Model-



model mismatch can be attributed to their different approaches and adopted observations. For example, CSIF uses broadband reflectances as indicators whereas downscaled TROPOMI SIF product benefits from TROPOMI's wider and denser spatial coverage. Nevertheless, CSIF-based $\alpha$ values (mostly > 20 $(\mu mol\,m^{-2}\,s^{-1})\,(mWm^{-2}\,sr^{-1}\,nm^{-1})^{-1}$) are higher than TROPOMI-based $\alpha$ of 18.5 $(\mu mol\,m^{-2}\,s^{-1})\,(mWm^{-2}\,sr^{-1}\,nm^{-1})^{-1}$ reported in Turner et al. (2020), which compensates for the SIF mismatch.

We also relate CSIF to vegetated and impervious fractions with a spatial resolution of 10 km from the World Urban Database and Access Portal Tools (WUDAPT) project (Ching et al., 2018) for a few available US cities. WUDAPT is "an international community-generated urban canopy information and modeling infrastructure to facilitate urban-focused climate, weather, air quality, and energy-use modeling application studies" (Ching et al., 2018). The Local Climate Zones (LCZ) provided by

WUDAPT contain 10 specific classifications for the urban areas and 7 natural types with characterizations of surface property/structure (e.g., building and tree height and density) and surface cover (pervious vs. impervious) (Stewart and Oke, 2012). Each LCZ classification is associated with a range of fractions for impervious land, building, and vegetation, such as 40 – 60 % of impervious percentage for compact high rise (Ching et al., 2018). We calculated the mean impervious and vegetated fractions for every LCZ and projected those fractions to CSIF's grids. Consequently, spatial gradient of CSIF

coincides with that of vegetated fractions estimated from WUDAPT, as suggested by the increasing trend moving away from urban cores (**Supplementary Fig. S3**). Thus, CSIF nicely reveals the urban-rural contrast in biological activities.

**Appendix B. Technical note on $R_{eco}$ prediction**

As introduced in **Sect. 2.3**, we have tested three ways to predict $R_{eco}$ by applying

M1)    one NN model trained using data points with all biomes to 0.05° grids, except for water, ice, and barren land, without
considering sub-gridcell variations in land cover types;

M2)    one NN model with biome and month and season of the year as additional categorical variables (with one-hot encoding);

M3)    multiple biome-specific NN models to 500 m grids with different biomes and aggregating predicted $R_{eco}$ to 0.05°.

Technically speaking, we tried both the built-in R package Neuralnet (Fritsch et al., 2016) as well as the R interface for Keras
(Falbel et al., 2019) that is a high-level neural networks API based on backend including TensorFlow to build NN models. Although M1 benefits from larger amount of sample data and shorter processing time in predicting $R_{eco}$ without matching NN models with land cover types at 500 m, M2 approach is physically more meaningful as the temperature dependence of respiration may vary among different vegetation species. In particular, being able to resolve sub-gridcell variations and different vegetation fractions (processed during GPP estimates, **Sect. 2.2.2**) facilitates a more reasonable $R_{eco}$ calculation over

urban areas. Note that modeled $T_{air}$ and $T_{soil}$ have been bilinearly interpolated from their initial grid spacings onto the required grid spacing. In the ends, we used the last M3 approach given its better performance.



Before training neural network models, we removed a few flux sites with relatively large RMSE values between modeled versus observed GPP to avoid erroneous information being transformed into the training process of R$_{eco}$ (**Supplementary Fig. S11**). All explanatory and response variables have been linearly normalized to values between [0, 1] based on their maximum and minimum values before training. Adjusting the raw data to a common scale may dramatically speed up the training process

and avoid the situation where the predicting variable appears to be more sensitive to one of the response variables. To determine the hyper-parameters that work the best on our data and help avoid under-/over-fitting situation, we carried out a 5-fold cross-validation (**Supplementary Fig. S12**). That being said, 80 % of the data serves as the training set with the remaining 20 % as the validation set. The 5-fold cross validation can be viewed as repeated holdouts and the average errors will be calculated after five different holdouts. The main hyper-parameter tested here is the number of neurons (8, 16, 32, or 64 neurons per

layer) for a two-layer model. The overall RMSE and loss function between predicted and observed R$_{eco}$ on the validating set turn out to be similar among different numbers of neurons. In addition, loss functions on the training and validating sets appear to be similar, implying no strong sign of overfitting.

**Author contributions.** DW and JCL designed the study and DW is responsible for model scripts. JCL and HD offered constructive insights regarding the construction and development of SMUrF. JCL, TO and EAK contributed to the design of

model result presentation. VY and NP provided the urbanVPRM model outputs for LA for flux comparison with SMUrF. All authors contributed to the editing of the final manuscript.

**Competing interests.** The authors declare that they have no conflict of interest.

**Acknowledgement**

This work is based upon work supported by the National Aeronautics and Space Administration funding under grant

NNX15AI40G, NNX15AI41G, NNX15AI42G, 80NSSC19K0196, and 80NSSC19K0093. TO is supported by NASA grant NNX14AM76G, 80NSSC18K1307, and 80NSSC18K1313. The resources and support from CHPC at the University of Utah are gratefully acknowledged. VY and NCP acknowledge funding by the NASA OCO-2 Science Team, grant number 17-OCO2-17-0025. A portion of this research was carried out at the Jet Propulsion Laboratory, California Institute of Technology, under a contract with the National Aeronautics and Space Administration. We are grateful to Ken Davis and Kai Wu at the

Penn State University for sharing INFLUX data and providing constructive comments on this manuscript. We acknowledge Alex Turner at the Univ. of Washington for sharing the unpublished, downscaled TROPOMI SIF data over the entire CONUS from June to Aug 2018. We thank Yao Zhang at the Lawrence Berkeley National Laboratory for sharing the 0.05° CSIF fields and for helpful discussions. DW also appreciates assistance from Guannan Wei (Purdue Univ.) and Benjamin Fasoli (Univ. of Utah) on the construction and evaluation of Neural Network models and discussions with Liyin He (Caltech) on the C3/C4

partitioning.



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





Figure 1. A demonstration of SMUrF (upper flow chart) and description of data products and observations used (bottom table). The temporal coverage in the table indicates the years used in this study.

| Product Name | | Spatial Info | Temporal Info | Usage |
|---|---|---|---|---|
| **MODIS12Q1 IGBP** | | 500 m, global | Annual mean, 2010-2018 | Provide biome types |
| **CSIF** | | 0.05°, global | 4-day mean, 2000-2018 | Estimate 4-day mean GPP |
| **FLUXNET2015** | | Global sites | Only use data from 2010-2014 | Calculate GPP-SIF slopes and train $R_{eco}$ |
| **MapSPAMv1** (physical crop areas) | | 10 km, global | Annual mean for 2010 | Estimate global C3/C4 crop ratios |
| **GlobBiomass** (above ground biomass) | | 100 m, global | Annual mean for 2010 | Estimate tree fractions in cities |
| **MOD44B** | | 1 km, global | Annual mean, 2010-2018 | Spatial downscaling (Optional) |
| **ERA5** | air and soil temperature | 0.25°, global | Daily mean, 2010-2018 | Predict $R_{eco}$ for the entire globe |
| | shortwave radiation | | Hourly mean, 2010-2018 | Hourly downscaling |
| **Daymet** (air temperature) | | 1 km, CONUS | Hourly mean, 2010-2018 | Predict $R_{eco}$ for CONUS |
| **NLDAS** (soil temperature) | | 0.125°, CONUS | Hourly mean, 2010-2018 | |
| **SMUrF** | | 0.05° | Hourly mean, 2010-2018 | Provide biospheric fluxes for cities |



**Figure 2.** Maps of derived C3-C4 ratio (yellow-green) based on physical areas of 42 crop species from the SPAMv1 model and weighted mean GPP-CSIF slopes for crops (purple-yellow) for the entire globe (left panel) and CONUS (right panel).



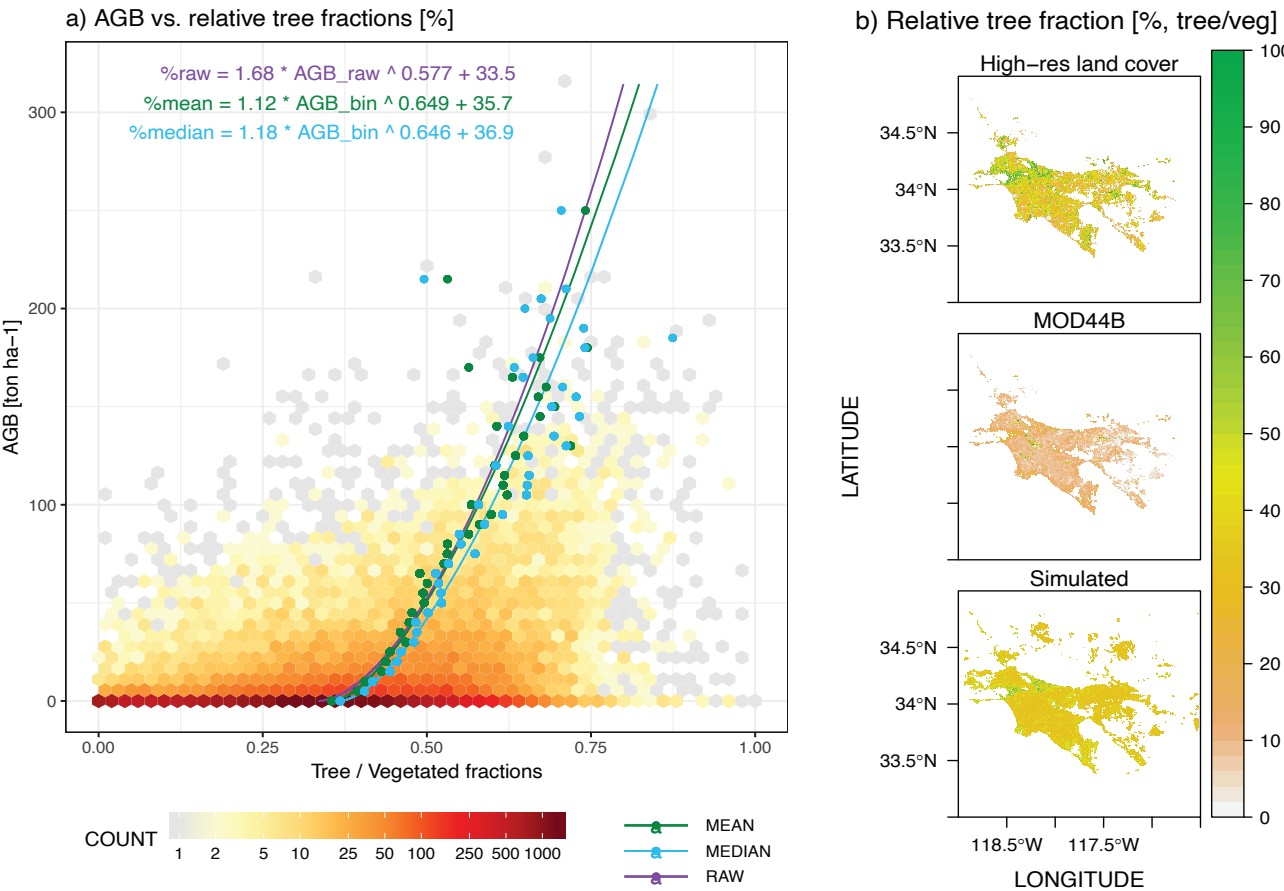

**Figure 3. a)** Power law relationships fitted between AGB and unbinned raw relative tree fractions (purple line) as well as between binned AGB and mean/median relative tree fractions per AGB bin (green/blue lines). **b)** Maps of calculated relative tree fractions (%) from a high-resolution land cover classification product (Coleman et al., 2020), MOD44B product, and our approximation based on AGB with a demonstration in panel a).



**Figure 4.** Estimated relative DBF (1st row) and non-tree fractions (i.e., GRA or OSHR, 2nd row) at 500 m and GPP-SIF slopes after urban gap-fills (3rd row) for Los Angeles (a), Chicago (b), and Boston (c).



**Figure 5.** Comparisons between directly computed modeled fluxes and FLUXNET displayed in density plots with 50 bins. **a-b)** 4-day mean observed GPP and directly computed SIF-based GPP at 89 global EC sites (a) and 78 non-crop sites (b) from 2010 to 2014. **c-d)** Daily mean observed $R_{eco}$ only on the testing set (i.e., 20% of all data) and directly computed $R_{eco}$ based on $T_{air}$ + $T_{soil}$ + GPP from FLUXNET (c) OR $T_{air}$ + $T_{soil}$ from ERA5 and SIF-based GPP (d) at all global EC sites from 2010 to 2014. **e-g)** Similar to panels **c)** and **d),** but for model-data $R_{eco}$ comparison ONLY at EC sites in the US with additional validations using Daymet $T_{air}$, NLDAS $T_{soil}$, and SIF-based GPP (g).

Although GPP and $R_{eco}$ were trained and predicted separately per biome, we collected and present available data from all biomes here. For each panel, numbers of data points in each of the 50 bins are displayed in log10 scales as yellow-blue colours. Error statistics like mean bias, correlation coefficient, and RMSE are printed based on the available data pairs. 1:1 and OLS-based regression lines are displayed in solid and dashed lines. Lastly, RMSEs derived from these direct validations were used for assigning errors at 500 m grid cell (**Sect. 2.5**).





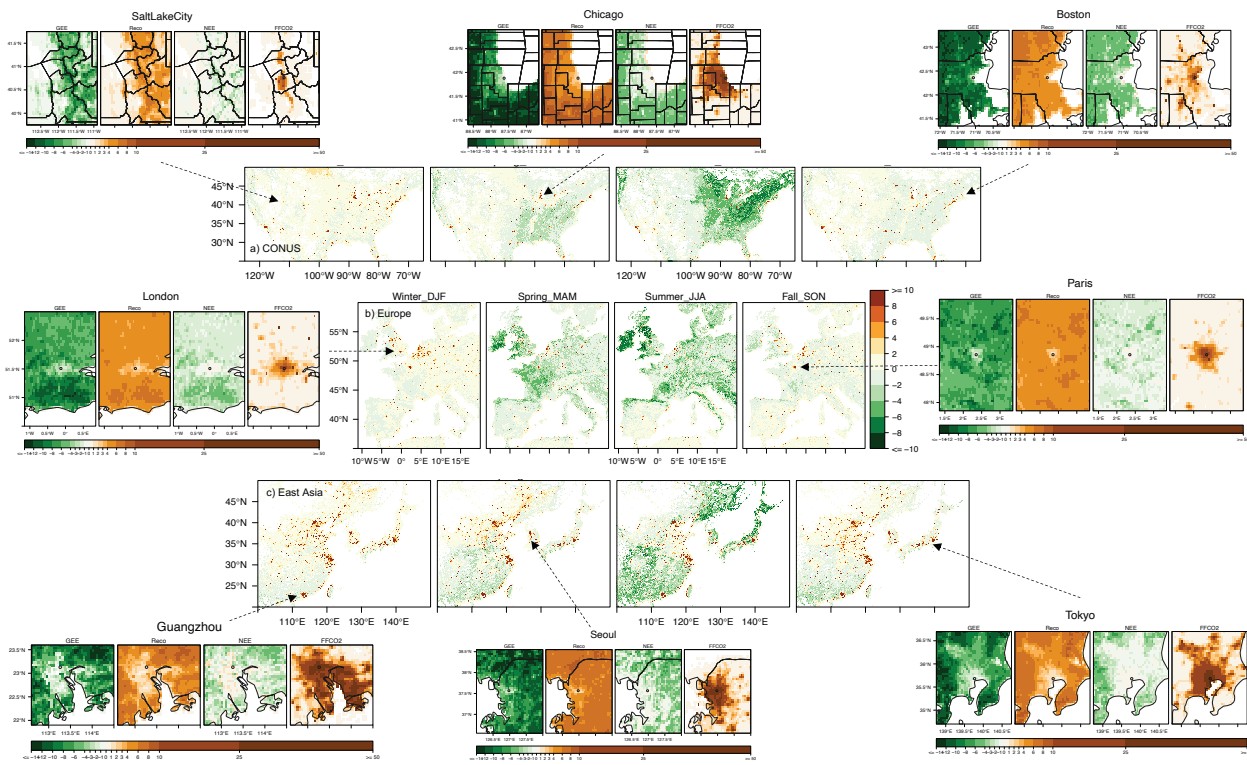

**Figure 6.** The sums of seasonal mean SMUrF-based NEE and ODIAC-based FFCO$_2$ ($\mu$mole m$^{-2}$ s$^{-1}$) for CONUS (a), western Europe (b), and East Asia (c) at 0.05º for 2018. Surrounded by regional maps in the center, spatial maps of summertime average (JJA 2018) GEE, R$_{eco}$, NEE, and FFCO$_2$ are zoomed into 8 cities (hereinafter **zoomed-in panels**).



**Figure 7.** A multi-city comparison of the spatially average NEE fluxes [$\mu$mol m$^{-2}$ s$^{-1}$] over urban **(a)** and rural **(b)** grid cells within a 2° × 2° area around the urban center from 2017 to 2018. We present the 4-day mean (in circles) and monthly mean (smoothed splines) NEE for 40 cities over CONUS, western Europe, east Asia, eastern Australia, parts of south America, and central Africa. Light grey ribbons indicate the typical north hemispheric summer months (June-Aug, JJA), so seasonal phases for Australian cities are the opposite from north hemispheric cities. The numbers printed below the city names denote the spatially average fossil fuel CO$_2$ emissions derived from ODIAC over the same grid cells as NEE.

## Mean Diurnal Cycle of NEE from SMUrF and FFCO2 from ODIAC over JJA 2018

○ FFCO2    △ NEE

### a) Urban grids within a 2° x 2° area

### b) Rural grids within a 2° x 2° area

**Figure 8.** A multi-city comparison of the average diurnal cycles of SMUrF-derived NEE fluxes (triangles and solid lines) and ODIAC/TIMES based FFCO₂ (circles) [$\mu$mol m$^{-2}$ s$^{-1}$] over JJA 2018 over the urban **(a)** and rural **(b)** grid cells within a 2° × 2° area around each urban center. *** Note that the y-scales of positive and negative fluxes are different for panel a) to better reveal the NEE fluxes, i.e., 0 to 60 $\mu$mol m$^{-2}$ s$^{-1}$ for positive fluxes and -20 to 0 $\mu$mol m$^{-2}$ s$^{-1}$ for negative fluxes. Light grey ribbons indicate the negative flux ranges from -10 to 0 $\mu$mol m$^{-2}$ s$^{-1}$ while light-yellow ribbons indicate the positive flux ranges from 0 to +10 $\mu$mol m$^{-2}$ s$^{-1}$. City names are labelled on the bottom of each panel.



Density plot of hourly mean NEE per biome for 67 EC sites in US and EU from 2010–2014
with 1:1 line (solid) and OLS fit (dashed)

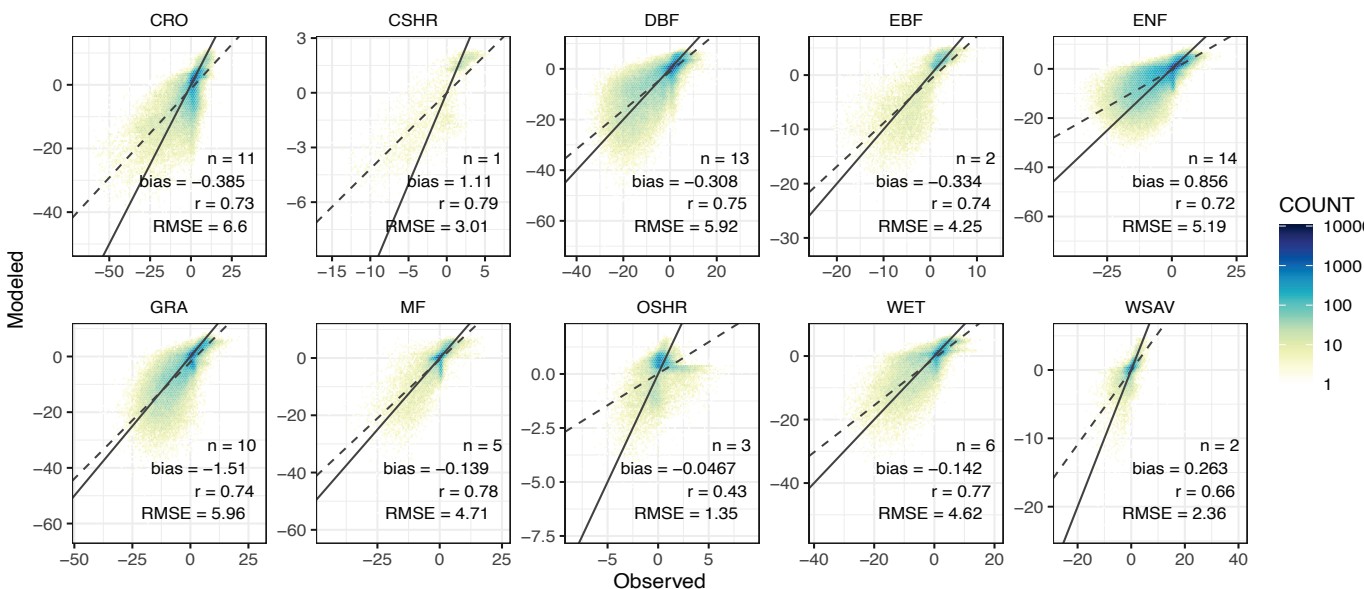

**Figure 9.** Hourly flux comparisons of ERA5-driven modeled NEE against measured NEE from FLUXNET presented in density plot for each biome with correlation coefficient labelled on the bottom.



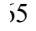

**Figure 10.** NEE flux evaluation against 4 EC towers from INFLUX around Indianapolis (**a**). Model-data flux comparisons [$\mu$mol m$^{-2}$ s$^{-1}$] from August 2017 to December 2018 are shown as density plot (**c**) and time series (**d**) using hourly mean NEE as well as monthly mean NEE (**e**). Model fluxes are shown in solid lines or circles with four colours indicating different sites, whereas measured fluxes are shown in black dashed lines or triangles. **f)** Mean diurnal cycle of observed NEE (black triangles) and modeled NEE (coloured circles) over JJA for two cropland sites (sites #2 and #3) and the entire year of 2018 for urban vegetation sites (sites #1 and #4).



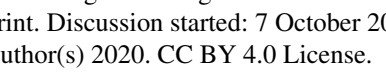

**Figure 11. a)** Spatial maps of mean GEE, $R_{eco}$, and NEE fluxes [$\mu mol\ m^{-2}\ s^{-1}$] from SMUrF and urbanVPRM over the greater Los
Angeles regions from July to September (JAS) in 2017. **b)** Mean diurnal cycles of NEE fluxes over JAS 2017 between two models.
**c)** Map of mean $FFCO_2$ [$\mu mol\ m^{-2}\ s^{-1}$] at 0.05° aggregated from 1 km ODIAC product over JAS 2017. Main cities have been labelled
on the maps including Los Angeles, Pasadena, Irvine, Riverside, and San Bernardino.



**Figure 12.** Demonstrations of the application of SMUrF in the context of column $CO_2$ measurements for an OCO-2 overpass on 1700UTC 7th July 2018 to the west of Boston. The time-integrated column footprints originated from dozens of receptor locations have been integrated along the latitude (**a**). The wind was blowing from the urban center of Boston onto the satellite swath (i.e., NNE wind). Spatial $XCO_2$ contributions from ODIAC-based $FFCO_2$ and SMUrF-based NEE ($XCO_{2.ff}$ and $XCO_{2.bio}$ in ppm) are plotted in panels **c)** and **d)**, respectively. Note that both spatial footprint (**panel a**) and $XCO_{2.ff}$ (**panel c**) are plotted in $\log_{10}$ scale and only relatively smaller contributions (< 1E-6 ppm) are displayed in light grey. Positive or negative $XCO_{2.bio}$ are plotted in green or orange colors (**d**). Lastly, the upwind contributions with respect to a column receptor are summed up to arrive at the total modeled $XCO_{2.ff}$ or $XCO_{2.bio}$ per receptor as orange or green dots in panel **b)**. Light orange, green, and blue ribbons indicate the urban, rural, and ocean latitude range, respectively.

**e)** A demo of incorporating $\Delta XCO_{2.bio}$ into the background estimates. The screened observed $XCO_2$ (gray triangles) are averaged in bins that match the modeled receptor (black triangles). The dark and light green lines imply the constant $XCO_2$ background and the bio-adjusted $XCO_2$ background. Differences between two $XCO_{2.bg}$ indicate the $\Delta XCO_{2.bio}$, which is calculated by subtracting the mean $XCO_{2.bio}$ within the rural range from all the simulated $XCO_{2.bio}$ along the swath.





**Figure 13.** Similar to **Figure 12a-d**, but for another three OCO-2 tracks, including 1800UTC 13[th] July 2018 to the east of Indianapolis (2018071318; 1[st] row, a-d), 2000UTC 25th June 2018 right over Salt Lake City (2018062520; 2[nd] row, e-h), and 1200UTC 15[th] June 2017 near Rome, Italy (2017061512; 3[rd] row, i-l). Meteorological fields that drove those simulations are 3km HRRR for US cities and 0.5degree GDAS fields for Rome.