# Peer review of "A Model for Urban Biogenic CO2 Fluxes: Solar-Induced Fluorescence for Modeling Urban biogenic Fluxes (SMUrF v1)"

_Geoscientific Model Development, 2020_

## Referee Comment (RC1) · Youngryel Ryu (Referee) · 22 Nov 2020

Dear Dien and all

I would like to congratulate you for this impressive manuscript. Incredibly comprehensive, in-depth analysis, great attentions to details and robust upscaling approach. I have to admit that I am not the expert in atmospheric transport model, where I didn't make any comment.

The authors developed a biogenic CO2 balance model which includes GPP, Reco, and NEE. They intended to develop this model for global cities, but actually it is applicable

to the global land. The basic idea came from linking SIF and GPP. They developed the slopes between GPP and SIF (CSIF products) across FLUXNET sites. After fine tuning (e.g. crops) in the slopes, they converted CSIF (0.05 degree) to GPP. As urban landscape is heterogeneous, they used very high resolution land cover maps to apply the slopes for the relevant land cover types then aggregated to 0.05 degree. Then the authors developed an Reco model using NN with GPP, Tair and Tsoil. To evaluate the model performance, the authors compiled FLUXNET, INFLUX dataset and rubanVPRM model. Then the authors combined fossil fuel emissions data, XCO2 data and an atmospheric transport model to tease out the contributions of biogenic CO2 fluxes in urban CO2 fluxes around the world.

The scope of this manuscript is vast but the authors didn't gloss over important details. Although some parts could be improved further, overall I see this is already too good. Though I would like to make some suggestions for furhter improvement.

First, evaluate SMUrF NEE directly against FLUXNET data like what you did for GPP and Reco in Fig 5. Good performance in GPP and Reco does not necessarily indicate good performance in NEE which is tiny signal compared to the other two fluxes. The authors reported Fig S10 for NEE evaluation, but I feel it is not enough. It is fine to report rather poor performance in NEE, which is quite well expected as machine learning based NEE (e.g. FLUXCOM) performed poorer than GPP and Reco. It would be an useful point about how to improve SMUrF later.

Second, the current evaluation focused on diurnal to seasonal scales. Could you provide some discussion on the model performance in interannual to trends? e.g. in case of LA, how NEE varied across dry and wet years? How does NEE/fossil fule CO2 varies across dry and wet years?

Third, I would like to recommend adding some discussions for including evaporation in SMUrF, not now but in v2. Your model already has most important components to compute evaporation. One approach would be to use Ball-Berry model to link

your GPP, canopy conductance and finally evaporation. I really enjoyed this paper (https://doi.org/10.1073/pnas.2005253117), which stressed the important linkage between irrigation and biogenic CO2 fluxes in LA. I think SMUrF can track this as well once evaporation module is included.

Followings include minor comments:

P4 L5-10: The previous paragraph criticized the limitation of simple Reco model, then this paragraph explained ML for SIF and land surface fluxes. I feel somewhat disconnected from the previous paragraph.

P9: pure temperature -> revise

P10 L30: I feel the assumption for no correlation between GPP and Reco is overly simplified. SMUrF model structure indicates GPP is a forcing to Reco (P6 L16).

P13 L6: What's GEE? Isn't it GPP?

P18 L12: what is QF?

P20 L3: spatial SIF -> revise

P20 L10-22: It is worth discussing complex SIF-GPP relationships reported in recent literature. Consistent, linear relationship disappears in some cases.

https://doi.org/10.1016/j.rse.2018.07.008      https://doi.org/10.1002/2017JG004180 https://doi.org/10.1038/s41598-018-32602-z

Again, this is a great manuscript. I really enjoyed reading it, and also learned a lot. Thanks- Youngryel

---

## Referee Comment (RC2) · Anonymous Referee #2 · 12 Jan 2021

This paper proposes a new model for estimating biogenic carbon fluxes from urban areas. This model, called SMUrF uses a new global solar-induced fluorescence product (cSIF) and biome specific GPP-SIF relationships to create a temporally and spatially explicit flux product specifically turned for urban vegetation, which is notoriously difficult to model accurately. Respiration is also carefully considered in this new model, using a neural net approach. The paper does an excellent job of describing the intricate (and very numerous) processes involved and the model, and the result is a truly exciting work that is sure to be of great interest to the flux modelling community. Although the SMUrF model contains a large number of assumptions (as any model of this scope does), and will likely be refined in the future, the authors cleverly acknowledge that this

is only the first iteration of the SMUrF model by referring to this version as "v1" in the main title.

The model relies on an assumption that GPP and SIF have a linear relationship, and that the slope of this relationship (alpha) is only a function of biome type. The plots for these calculations are buried in the supplemental, and for many of the biomes, the relationship does not appear to be linear. It is unclear if this non-linearity in addressed in the uncertainty analysis. This is the one part of the analysis that I wish was discussed in more detail.

In an effort to relate the SMUrF model output to XCO2 observations from OCO-2, the X-STILT transport model was used to generate total-column footprints. The assemblage of Boston area footprints shown in Figure 12a. shows a satellite overpass that occurred while the winds were out of the NNE along the flight track. The forward model results shown in Figure 12b seem ok, but the plots in 12b,c are confusing, because they are not XCO2, they are the spatially explicit contributions to the XCO2 concentrations for the satellite observations. The analysis in Figure 12e seems problematic, particularly for the treatment of the background concentration. The background value chosen appears somewhat arbitrary and taken from a region downwind of the city. The correlation between the binned OCO-3 observations (black triangles) and the full model result (purple line) is not particularly strong. The author states that the additional of SMUrF to the analysis is an improvement over just using a fossil fuel inventory, but other papers (such as the cited Sargent, 2018) spend a lot more time dealing with incorporating the biosphere with these types of transport models. While SMUrF represents an important step forward in assimilating SIF measurements into a biosphere carbon flux model, the STILT analysis at the end is incomplete, and, in my opinion, the paper would be better off for dropping this part entirely. Many researchers will surely be eager to explore the use of SMUrF with transport models to compare with satellite data, but these comparisons will need to spend a lot more time on dealing with subtleties such as determining the background. Because XCO2 anomalies are so small over cities (typically a few

ppm at most), a careful error analysis would also be needed, which is lacking here.

The manuscript contains a large number of figures, many with numerous subplots. While this isn't uncommon for GMD papers describing a new model, this particular work would benefit from slimming down some of the figures. I've discussed a few of the figures individually below:

Figure 1: This is a really well laid out flow chart. It took me a while to get through it all, but it was really helpful in understanding the model, and I like how it was labeled with section and figure references.

Figure 2: Subplot c needs units for alpha values. Also, subplots are not labeled.

Figure 6: This figure is way too complicated. In addition to their being too many cities, I can't easily discern what the take home message is supposed to be from all of these plots.

Figure 7: Again, too many subplots. It would be easier to read if thier were fewer cities selected. To me, the interesting information in this figure is both the magnitude of max NEE for different cities and the timing of when that max NEE occurs. Perhaps it would be more impactful to show a different type of plot. Perhaps a scatter plot with the x-axis being day-of-year for the NEE peak and the y-axis being peak magnitude? You could then pack a bunch more cities into one plot, and label the cities in the scatterplot.

Figure 8: These time of day plots are nice, but the half-circle makers are hard to see.

Figure 10: Again, too many panels.

Figure 11: This is great. I wish there was more urbanVPRM comparisons with other cities. A real test of the usefulness of SMUrF is its performance compared to other models, especially those also tailored for urban areas.

Overall, this is an impressive manuscript. The model described is sure to make an impact in the community, and I know that I and other researchers look forward to working

with it.

Specific line-by-line comments (mostly grammar stuff) are below:

p.6 L3 Grammar (". . .by trained on. . .")

p.6 L10 "Laser" capitalized

p.6 L12 punctuation

p.6 L28 Underline on part of "(Sect. 3.2)."

p.13 L34 "than" -> "rather than"

p.14 L26 Why? Please add a sentence of explanation.

p.15 L12 "prediction" -> "predictions"

p.15 L13 "as" -> "of"

p.15 L19 "amount" -> "amounts"

p.16 L15 "comparison" -> "comparisons"

p.16 L15 "insights on" -> "insight into"

p.16 L25 "turns" -> "turn"

p.16 L26 "GPP," -> "GPP as well as"

p.16 L31 Confusing sentence, please rewrite.

p.16 L32 "grids" -> "gridcells"

p.17 L11 "examine" -> "examined"

p.18 L25 "how" -> "how a"

p.18 L25 "bio-gradient" -> "gradient"

p.20 L6 "on-board" -> "onboard"

p.20 L28 "10" in "Q10" shouldn't be italicized

---

## Author Comment (AC1) · 1 Mar 2021

**Response to Referee #1**

We thank both referees for their efforts and constructive comments. Each referee's comments are shown below in *italics*, followed by our point-by-point responses in blue and relevant text in red.

**Referee #1 Youngryel Ryu (Referee) ryuyr77@gmail.com**

*Dear Dien and all*

*I would like to congratulate you for this impressive manuscript. Incredibly comprehensive, in-depth analysis, great attentions to details and robust upscaling approach. I have to admit that I am not the expert in atmospheric transport model, where I didn't make any comment.*

*The authors developed a biogenic $CO_2$ balance model which includes GPP, $R_{eco}$, and NEE. They intended to develop this model for global cities, but actually it is applicable to the global land. The basic idea came from linking SIF and GPP. They developed the slopes between GPP and SIF (CSIF products) across FLUXNET sites. After fine tuning (e.g. crops) in the slopes, they converted CSIF (0.05 degree) to GPP. As urban landscape is heterogeneous, they used very high resolution land cover maps to apply the slopes for the relevant land cover types then aggregated to 0.05 degree. Then the authors developed an $R_{eco}$ model using NN with GPP, Tair and Tsoil. To evaluate the model performance, the authors compiled FLUXNET, INFLUX dataset and urbanVPRM model. Then the authors combined fossil fuel emissions data, $XCO_2$ data and an atmospheric transport model to tease out the contributions of biogenic $CO_2$ fluxes in urban CO2 fluxes around the world.*

*The scope of this manuscript is vast but the authors didn't gloss over important details. Although some parts could be improved further, overall I see this is already too good. Though I would like to make some suggestions for further improvement.*

We truly appreciate the recognition and constructive comments from referee #1 Youngryel Ryu and have tried our best to conduct additional analyses and to improve the presentation of this manuscript.

*First, evaluate SMUrF NEE directly against FLUXNET data like what you did for GPP and $R_{eco}$ in Fig 5. Good performance in GPP and $R_{eco}$ does not necessarily indicate good performance in NEE which is tiny signal compared to the other two fluxes. The authors reported Fig S10 for NEE evaluation, but I feel it is not enough. It is fine to report rather poor performance in NEE, which is quite well expected as machine learning based NEE (e.g. FLUXCOM) performed poorer than GPP and $R_{eco}$. It would be an useful point about how to improve SMUrF later.*

We agree that NEE evaluation against observations is critical. In the initial manuscript, we have already included a model evaluation of the HOURLY mean biome-specific NEE fluxes against FLUXNET data (Fig. 9 and Sect. 3.2.1). In accordance with the reviewer's point, NEE has a poorer performance compared to GPP and $R_{eco}$, especially for biomes with less training data (i.e., FLUXNET observations).

*Second, the current evaluation focused on diurnal to seasonal scales. Could you provide some discussion on the model performance in interannual to trends? e.g. in case of LA, how NEE varied across dry and wet years? How does NEE/fossil fuel $CO_2$ varies across dry and wet years?*

While the paper would be more informative with additional discussion on the interannual variation/trend in urban NEE, the primary motivation of developing the model is to separate the anthropogenic and biogenic $CO_2$ signals when interpreting the atmospheric measurements. The column $CO_2$ retrieved from satellites may be affected by the upstream fluxes over timescales of only O(day). Therefore, we argue that $CO_2$ fluxes at the diurnal scale matters the most to the downwind atmospheric $CO_2$ observations than $CO_2$ fluxes at much longer timescales. For example, the interannual variation/trend in NEE may be a secondary-order effect, superimposed on the urban-rural difference in NEE fluxes that influence the $CO_2$ background determination. Finally, the paper

is already quite lengthy, and considering the main motivation of the manuscript, we are inclined not to touch on NEE fluctuation at the moment, but added relevant text in the future work section for clarification (Sect. 4.2) with changes highlighted in red:

> "Because atmospheric $CO_2$ concentrations measured from satellites are mainly influenced by the anthropogenic and biogenic $CO_2$ fluxes a few hours to days ahead of the overpass time, this work focused on presenting and evaluating the diurnal and seasonal $CO_2$ fluxes. Biogenic $CO_2$ fluxes over longer timescales, e.g., their interannual variations and long-term trends, may require further investigations. We also hope to examine more cities and different times of the day in future studies to better study the relative biogenic and anthropogenic contributions to $XCO_2$ anomalies. And, incorporating uncertainties in biogenic fluxes and resultant $XCO_{2.bio}$ is needed for future studies with aims of understanding urban signals especially over the growing seasons."

*Third, I would like to recommend adding some discussions for including evaporation in SMUrF, not now but in v2. Your model already has most important components to compute evaporation. One approach would be to use Ball-Berry model to link your GPP, canopy conductance and finally evaporation. I really enjoyed this paper (https://doi.org/10.1073/pnas.2005253117), which stressed the important linkage between irrigation and biogenic $CO_2$ fluxes in LA. I think SMUrF can track this as well once evaporation module is included.*

> We thank the reviewer for the suggestion and agree that an evaporation module is a great function to be implemented in the future. Urban irrigation plays an important role in GPP especially for semiarid urban areas, (e.g., Loridan et al., 2010, Johnson and Belitz 2012, Vahmani and Hogue 2014, Miller et al., 2021). We added the following discussions in Sect. 4.2:

>> "Since carbon fluxes are closely tied to the water cycle, anthropogenic moisture input (i.e., urban irrigation) can effectively influence the urban biogenic fluxes, particularly over arid and semi-arid residential areas like Salt Lake City and Los Angeles (Loridan et al., 2010, Johnson and Belitz 2012, Litvak et al., 2017, Miller et al., 2021). Although we currently rely on SIF to pick up potential irrigation effect on urban GPP, it is possible and informative to further relate the water flux exchange to the carbon flux exchange in the future version of SMUrF."

**Followings include minor comments:**

*P4 L5-10: The previous paragraph criticized the limitation of simple $R_{eco}$ model, then this paragraph explained ML for SIF and land surface fluxes. I feel somewhat disconnected from the previous paragraph.*

> We thank the reviewer for pointing out the disconnection. Our initial thought is to highlight the difficulties and limits in estimating $R_{eco}$, which is also the motivation of adopting ML technique for $R_{eco}$ estimates. We have now improved the flow of the two paragraphs (with modified text in red):

>> "...After all, the complexity of biological and non-biological processes of $R_{eco}$ and the lack of mechanistic understanding of how biotic and abiotic factors affect $R_{eco}$ make the mechanistic modeling quite challenging. Given the complexity in modeling $R_{eco}$, machine learning (ML) techniques will be adopted in this study.

>> ML approaches have been increasingly applied in many disciplines including ecosystem modeling to help answer complicated, entangled problems through extracting patterns from data streams for predictions and generalizations..."

*P9: pure temperature -> revise*

> We have now changed "pure temperatures and GPP observations" to "direct temperatures and GP observations".

*P10 L30: I feel the assumption for no correlation between GPP and $R_{eco}$ is overly simplified. SMUrF model structure indicates GPP is a forcing to $R_{eco}$ (P6 L16).*

We agree that the assumption for no correlation is overly simplified and have added a correlation term when calculating the uncertainties in NEE in the code.

*P13 L6: What's GEE? Isn't it GPP?*

GEE is the gross ecosystem exchange and closely related to GPP. GEE is often used by researchers working with eddy covariance observations, while GPP is used more often by ecologists. We define GEE = -GPP, so that photosynthetic uptake represents removal from the atmosphere (Fig. 6).

*P18 L12: what is QF?*

QF stands for quality flag, a measure provided by the OCO-2 $XCO_2$ retrieval to indicate data quality. We have now modified that sentence.

*P20 L3: spatial SIF -> revise*

We have now changed "spatial SIF over cities" to "SIF retrieval with a broader spatial coverage over cities".

*P20 L10-22: It is worth discussing complex SIF-GPP relationships reported in recent literature. Consistent, linear relationship disappears in some cases. https://doi.org/10.1016/j.rse.2018.07.008 https://doi.org/10.1002/2017JG004180 https://doi.org/10.1038/s41598-018-32602-z*

We thank the reviewer for sharing these relevant publications and have added some text on the nonlinearity of GPP-SIF relationship in the discussion section (Sect. 4.2):

"Second, GPP within SMUrF is currently estimated as a linear function of SIF, using a set of constant biome-specific linear slopes (α) without considering temporal or inter-site variations. The adoption of SIF has dramatically benefited and simplified the GPP calculation, as no extra satellite indices or impervious fractions need to be plugged in. However, previous research based on ground-based SIF measurements (Miao et al., 2018; Wohlfahrt et al., 2018; Yang et al., 2018) revealed the GPP-SIF relation deviated from linearity at the sub-diurnal scale, under unstable light conditions, or heat stress. While SIF and absorbed PAR are linearly related, the GPP-SIF relationship can deviate from linearity due to complex LUE:SIF yield relationships in light-saturating vs. light-limiting regimes (Miao et al., 2018). Thus, considering additional environmental factors related to the modeling of light use efficiency—e.g., relative humidity, cloudiness, and growth stage of crops, could improve the SIF-based GPP estimates (Yang et al., 2018). Although the nonlinear GPP-SIF relationship was not explicitly accounted for in this first iteration of SMUrF, our estimated flux uncertainties against dozens of flux tower sites implicitly account for the overall potential error associated with the linear assumption. Nevertheless, we anticipate future efforts to add more degree of freedoms in the estimate of GPP-SIF relation."

*Again, this is a great manuscript. I really enjoyed reading it, and also learned a lot.*
*Thanks- Youngryel*

Thank you, Youngryel. – Dien on behalf of the team

---

## Author Comment (AC2) · 1 Mar 2021

**Response to Referee #2**

We thank both referees for their efforts and constructive comments. Each referee's comments are shown below in *italics*, followed by our point-by-point responses in blue and relevant text in red.

**Anonymous Referee #2**

*This paper proposes a new model for estimating biogenic carbon fluxes from urban areas. This model, called SMUrF uses a new global solar-induced fluorescence product (cSIF) and biome specific GPP-SIF relationships to create a temporally and spatially explicit flux product specifically turned for urban vegetation, which is notoriously difficult to model accurately. Respiration is also carefully considered in this new model, using a neural net approach. The paper does an excellent job of describing the intricate (and very numerous) processes involved and the model, and the result is a truly exciting work that is sure to be of great interest to the flux modelling community. Although the SMUrF model contains a large number of assumptions (as any model of this scope does), and will likely be refined in the future, the authors cleverly acknowledge that this version Discussion paper is only the first iteration of the SMUrF model by referring to this version as "v1" in the main title.*

We appreciate the constructive feedback from the reviewer and have tried our best to improve the clarity of the text plus figures and redo the demonstration on analyzing column $CO_2$ observations.

*The model relies on an assumption that GPP and SIF have a linear relationship, and that the slope of this relationship (alpha) is only a function of biome type. The plots for these calculations are buried in the supplemental, and for many of the biomes, the relationship does not appear to be linear. It is unclear if this non-linearity in addressed in the uncertainty analysis. This is the one part of the analysis that I wish was discussed in more detail.*

We are aware of the non-linearity between GPP and SIF at finer temporal scales (e.g., sub-diurnal scales), suggested by a few studies analyzing the ground-based SIF and GPP measurements. The non-linearity results from the complex relationship between the light use efficiency (GPP / APAR) and the SIF yield (SIF / APAR) and its non-linear behavior under different light conditions. Although the

We added two discussions on the nonlinearity of GPP-SIF relationship in the methodology section (Sect. 2.5):

"It is worth noting that non-linearity in GPP and SIF has been reported under some circumstances, e.g., sub-diurnal scales or unstable light conditions (e.g., Yang et al., 2018, Miao et al., 2018). The uncertainties in assuming linear GPP-SIF relationships across biomes were not explicitly quantified but were implicitly accounted for as part of the total uncertainties quantified from the model-observation comparison."

as well as in the discussion section (Sect. 4.2):

"Second, GPP within SMUrF is currently estimated as a linear function of SIF, using a set of constant biome-specific linear slopes ($\alpha$) without considering temporal or inter-site variations. The adoption of SIF has dramatically benefited and simplified the GPP calculation, as no extra satellite indices or impervious fractions need to be plugged in. However, previous research based on ground-based SIF measurements (Miao et al., 2018; Wohlfahrt et al., 2018; Yang et al., 2018) revealed the GPP-SIF relation deviated from linearity at the sub-diurnal scale, under unstable light conditions, or heat stress. While SIF and absorbed PAR are linearly related, the GPP-SIF relationship can deviate from linearity due to complex LUE:SIF yield relationships in light-saturating vs. light-limiting regimes (Miao et al., 2018). Thus, considering additional environmental factors related to the modeling of light use efficiency—e.g., relative humidity, cloudiness, and growth stage of crops, could improve the SIF-based GPP estimates (Yang et al., 2018). Although the nonlinear GPP-SIF relationship was not explicitly accounted for in this first iteration of SMUrF, our estimated flux uncertainties against dozens of flux tower sites implicitly account for the overall potential error associated with the linear assumption. Nevertheless, we anticipate future efforts to add more degree of freedoms in the estimate of GPP-SIF relation."

*In an effort to relate the SMUrF model output to XCO$_2$ observations from OCO-2, the X-STILT transport model was used to generate total-column footprints. The assemblage of Boston area footprints shown in Figure 12a. shows a satellite overpass that occurred while the winds were out of the NNE along the flight track. The forward model results shown in Figure 12b seem ok, but the plots in 12b,c are confusing, because they are not XCO$_2$, they are the spatially explicit contributions to the XCO$_2$ concentrations for the satellite observations.*

Figure 12c,d show the anthropogenic and biogenic contributions in ppm from each upwind grid cell with respect to the downwind XCO$_2$. As mentioned in the figure caption, we often referred to those anthropogenic and biogenic contributions as spatial XCO$_{2.ff}$ and XCO$_{2.bio}$. These spatial contributions have units of ppm and are further calculated from the product between X-STILT column footprint [ppm /(μmol m-2 s-1)] and upwind fluxes [μmol m-2 s-1]. The spatial sum of these contributions arrives at the total anthropogenic and biogenic anomalies at corresponding receptors, as shown in panel b).

*The analysis in Figure 12e seems problematic, particularly for the treatment of the background concentration. The background value chosen appears somewhat arbitrary and taken from a region downwind of the city. The correlation between the binned OCO-3 observations (black triangles) and the full model result (purple line) is not particularly strong. The author states that the additional of SMUrF to the analysis is an improvement over just using a fossil fuel inventory, but other papers (such as the cited Sargent, 2018) spend a lot more time dealing with incorporating the biosphere with these types of transport models. While SMUrF represents an important step forward in assimilating SIF measurements into a biosphere carbon flux model, the STILT analysis at the end is incomplete, and, in my opinion, the paper would be better off for dropping this part entirely. Many researchers will surely be eager to explore the use of SMUrF with transport models to compare with satellite data, but these comparisons will need to spend a lot more time on dealing with subtleties such as determining the background. Because XCO$_2$ anomalies are so small over cities (typically a few ppm at most), a careful error analysis would also be needed, which is lacking here.*

1) We agree with the reviewer's criticism - it might be too soon to conclude that using SMUrF can effectively improve the background along satellite swath, particularly given only one case examined. However, we hope to provide a demonstration of how SMUrF can be used with transport models and emphasis the role of urban-rural gradient in NEE and resultant biogenic XCO$_2$ signals played in the background definition. We feel that even though this is a model description paper, it is illustrative for the reader to see an application of the model in helping to interpret satellite XCO$_2$ data.

   **To deemphasize the quantitative results from a limited number of analyses, we modified a relevant sentence in the Abstract to read.**

   Initial text - "By examining a few summertime satellite tracks over four cities, we found that the urban-rural gradient in column CO$_2$ (XCO$_2$) anomalies due to NEE can sometimes reach ~0.5 ppm and be close to XCO$_2$ enhancements due to FFCO$_2$ emissions."

   **Modified text - "To illustrate the application of SMUrF, we used it to interpret a few summertime satellite tracks over four cities and compared the urban-rural gradient in column CO$_2$ (XCO$_2$) anomalies due to NEE against XCO$_2$ enhancements due to FFCO$_2$ emissions."**

2) In terms of the background definition, we agree that the initial choice can be arbitrary and calculating the proper background for column data is especially challenging and has been extensively investigated in our previous work (Wu et al., 2018, hereinafter Wu2018). **We now followed the overpass-specific approach illustrated in Wu2018, that is calculated from the average observed XCO$_2$ over the background latitude band.** The relevant figure regarding the overpass-specific approach has been added to the supplement as Fig. S13 in the revised manuscript.

Specifically, we leveraged the forward-mode of STILT where STILT particles were continuously released forward in time from a box around Boston for 12 hours. The border of the urban plume is defined by fitting a normalized 2D kernel density (purple contour in the **Fig. S13** to the right) to particle locations during the few minutes where OCO-2 overpass the city. The intersection of the urban plume and the satellite swath give rise to the urban-polluted latitude range (red triangles in panel b). Then the background latitude range (42.26 to 42.76ºN, green ribbon in panel b) is chosen to the north of the urban-polluted range, given 1) the geometry between the swath orientation and the wind direction and 2) possible contamination from oceanic fluxes over the region to the south of the urban plume.

We arrived at both the mean background along with its uncertainty for this swath as shown in dotted-dashed green line and green ribbon, which have values of 403.37 +/- 1.03 ppm. The background uncertainty can also be used in error analyses and atmospheric inversions like those conducted in Wu2018.

Supplementary Figure S13. Overpass-specific background following Wu et al. (2018).

[Figure]

3) **Next, we explain the reason for using this overpass-specific background**:

Wu2018 has carefully invested three common methods with different complexity in estimating $XCO_{2,bg}$. The reviewer is welcomed to read over Sect. 2.3 and Sect. 3.3 in Wu2018 for full details. For the convenience, we summarized main messages as follows and adopted the relevant two figures from Wu2018 (Figure S8 and S12, also shown on the next page) for explanations:

M1. a "trajectory-endpoint" method is investigated by assigning $CO_2$ values extracted from global models (e.g., CarbonTracker, CT) to trajectory endpoints including simulating biospheric, oceanic, and prior components. This method has been widely used in Lagrangian-based modeling work including the cited Sargent et al., 2018. **However, most prior work only had to deal with $CO_2$ measurements within the PBL where huge $CO_2$ anomalies are caused by either anthropogenic or biogenic. When applying this trajectory-endpoint method solely relying on model simulations to interpreting column $CO_2$ measurements, this approach may often lead to potential "bias" in background values (orange lines in Figure S12 adopted from Wu2018).** Although there is hardly a "truth" for $XCO_2$ background, the modeled total $XCO_2$ based on this approach appears to be unreasonably higher or lower than the retrieved $XCO_2$ by 1-2 ppm (orange dotted-dashed line vs. black solid line in Figure S8 of Wu2018). These mismatches of 1-2 ppm can already be huge given small anthropogenic $XCO_2$ enhancements and is caused by potential uncertainties in the adopted global models (e.g., CT) with accumulated errors in the endpoint of STILT (further result from wind errors).

M3. an "overpass-specific" background as described earlier. **$XCO_{2,bg}$ calculated by combining observations and wind information from forward-mode of STILT can be more consistent with the retrieval and account for upwind-downwind geometry, which is better than approaches that solely rely on models OR observations. The biggest hurdle would be wind bias in STILT, which unfortunately can affect the M1 trajectory-endpoint background as well.**

[Figure]

*The following Figure S12 and Figure S8 are adopted from Wu et al. (2018).*

**Figure S12.** Same as **Fig. 6e**, except for using OCO-2 Lite b8. Numbers labeled in darkgreen donote the amount of screened soundings (QF = 0) using b8 in the background. Due to only 8 soundings for overpass on 2014122910, background uncertainty is hard to estimate (no error bar displayed).

[Figure]

**Figure S8.** Same as **Fig. 8**, but for all five overpasses examined over Riyadh using OCO-2 Lite v7.

4) Lastly, we agree with the reviewer that the biogenic adjusted signal may not correlate strongly with the observations, possibly due to various reasons, e.g., 1) the bias in near-field wind direction, 2) uncertainties in both FF and biogenic fluxes, 3) retrieval error. We have now added a wind error analysis by comparing the modeled wind speed and directions against a NOAA radiosonde station (41.67N, 69.97W) adjacent to Boston city. Close to the overpass hour (07/07/2018 17 UTC), we see overall positive biases in the HRRR-based wind direction from the surface to 3 km. This positive bias potentially explains a northward modeled $XCO_2$ peak than the observed peak (latitude shift of about 0.1 degreeN in **Fig. 12e**).

| time.string | u.bias | v.bias | ws.bias | wd.bias | rmse |
|---|---|---|---|---|---|
| 07/07/2018 00 UTC | -2.29 | -1.86 | -1.36 | 16.6 | 3.08 |
| 07/07/2018 12 UTC | 0.220 | 1.22 | -1.70 | 2.68 | 1.91 |

Revised Figure 12e:

[Figure]

We may argue that if the aim of a study is to quantify FF emissions over cities surrounded by vegetations (e.g., Sargent et al., 2018), a comprehensive error analysis or even an atmospheric inversion is needed. In future work, we will follow the full error analysis and potentially a scaling factor type atmospheric inversion conducted in Wu2018 to make more quantitative results. However, given the already lengthy manuscript and the main scope of this work being model presentation, we simply modified the text in Sect 4.1 for clarifications.

"To facilitate visualization and understanding of $\Delta XCO_{2.bio}$ and bio-adjusted background, let us return to the Boston case again (**Fig. 12**). Following the "overpass-specific approach" proposed in Wu et al. (2018), we took the near-field wind direction into account and defined the background latitude range as 42.26°–42.76° N (light green ribbon in **Supplementary Fig. S13b** and **Fig. 12e**). The constant background is 403.37 ppm (dark green line in **Fig. 12e**) with an uncertainty of 1.03 ppm containing both the retrieval uncertainty and $XCO_2$ noise in the background range. The mean $XCO_{2.ff}$ and $XCO_{2.bio}$ anomalies within the background region are 0.23 ppm and −1.41 ppm, respectively. After integrating the bio-gradient $\Delta XCO_{2.bio}$, a new bio-adjusted background varies along latitude (light green line in **Fig. 12e**). If modeled $XCO_{2.ff}$ is added to the bio-adjusted background, the resultant total $XCO_2$ better reproduces the latitudinal variations of the measured mean values (**Fig. 12e**). Both the observed $XCO_2$ and modeled $XCO_2$ correcting for $\Delta XCO_{2.bio}$ exhibit dips in $XCO_2$ on both sides outside the urban peak, which is missing from the model result using the constant background (orange line in **Fig. 12e**).

A comprehensive error analysis is required in future work to draw quantitative conclusions from model-data $XCO_2$ comparisons given various uncertainty sources. For instance, the modeled $XCO_2$ appears to be broader latitudinally with a lower amplitude and a small latitude shift of around 0.1° compared to observed $XCO_2$ (purple line versus black triangles in **Fig 12e**) likely due to bias in wind speed and direction. Nonetheless, neglecting the latitudinal/spatial gradient in biogenic $XCO_2$ anomalies given gradients in NEE affects the extracted urban signal and the inferred $FFCO_2$ emissions in this case."

We also reemphasis the future needs in Sect. 4.2 and deemphasis the quantitative conclusion in the abstract (as mentioned in above point 1):

"We also hope to examine more cities and different times of the day in future studies to better study the relative biogenic and anthropogenic contributions to $XCO_2$ anomalies. And, incorporating uncertainties in biogenic fluxes and resultant $XCO_{2.bio}$ is needed for future studies with aims of understanding urban signals especially over the growing seasons."

In summary, we clarify the key point in this manuscript being the urban-rural gradient in biogenic fluxes and $CO_2$ anomalies. Even though one may not follow the same exact constant background approach with biogenic adjustments as we showed in Sect. 4.1, one needs to consider the urban-rural contrast in biogenic fluxes that is lacking in many $XCO_2$-based studies.

*The manuscript contains a large number of figures, many with numerous subplots. While this isn't uncommon for GMD papers describing a new model, this particular work would benefit from slimming down some of the figures. I've discussed a few of the figures individually below:*

We appreciate these individual comments and have made some rearrangements to the figures.

*Figure 1: This is a really well laid out flow chart. It took me a while to get through it all, but it was really helpful in understanding the model, and I like how it was labeled with section and figure references.*

We thank the reviewer for the recognition and are very glad this flow chart worked well in the end.

*Figure 2: Subplot c needs units for alpha values. Also, subplots are not labeled.*

We have now added the unit for $\alpha$ values in the figure caption, i.e., (umol m$^{-2}$ s$^{-1}$) / (mW m$^{-2}$ nm$^{-1}$ sr$^{-1}$).

*Figure 6: This figure is way too complicated. In addition to their being too many cities, I can't easily discern what the take home message is supposed to be from all of these plots.*

We may argue that this zoom-in-and-out panel plot gives a nice overview of 1) anthropogenic and biogenic $CO_2$ fluxes from urban center to its surrounding and 2) how regional total $CO_2$ fluxes vary with seasons. One can spot the urban hotspots where anthropogenic $CO_2$ "beat" down the biogenic $CO_2$ fluxes.

*Figure 7: Again, too many subplots. It would be easier to read if thier were fewer cities selected. To me, the interesting information in this figure is both the magnitude of max NEE for different cities and the timing of when that max NEE occurs. Perhaps it would be more impactful to show a different type of plot. Perhaps a scatter plot with the x-axis being day-of-year for the NEE peak and the y-axis being peak magnitude? You could then pack a bunch more cities into one plot, and label the cities in the scatterplot.*

We appreciate the suggestions from the reviewer but may argue that the current presentation can provide a broad view for cities across multiple continents and hopefully facilitate readers with different cities of interests.

*Figure 8: These time of day plots are nice, but the half-circle makers are hard to see.*

We have now replaced circles with solid dots.

*Figure 10: Again, too many panels.*

We have now moved two of the initial six panels to the supplement (now as Fig. S12).

*Figure 11: This is great. I wish there was more urbanVPRM comparisons with other cities. A real test of the usefulness of SMUrF is its performance compared to other models, especially those also tailored for urban areas.*

> Yes – we also wish to provide more model comparisons with other observations and model products for more locations, which require additional support and collaboration from data providers/users in the field.

*Overall, this is an impressive manuscript. The model described is sure to make an impact in the community, and I know that I and other researchers look forward to working with it.*

> We thank the support from the reviewer and will keep improving the SMUrF model, given increasing understanding towards SIF, respiration, and urban biosphere as well as the availability of upcoming remote sensing data.

**Specific line-by-line comments (mostly grammar stuff) are below:**

*p.6 L3 Grammar (". . .by trained on. . .")*

> Corrected – changed 'by trained on' to 'that trained on'

*p.6 L10 "Laser" capitalized*

> Corrected – "Laser" to "laser".

*p.6 L12 punctuation*

> We've modified this sentence as "AGB and its grid-level uncertainty [tons ha$^{-1}$] by definition describe the "oven-dry weight of the…".

*p.6 L28 Underline on part of "(Sect. 3.2)."*
*p.13 L34 "than" -> "rather than"*

> Corrected.

*p.14 L26 Why? Please add a sentence of explanation.*

> The point where NEE becomes negative is the number we read from Figure 3A3 and 3B3 in Hardiman et al. 2017 (also attached on the bottom right). They show that the NEE turned negative at ~5 am local time in July 2013. We extract SMUrF fluxes from the similar Boston area considered in Hardiman2017 and calculated the monthly mean diurnal cycle with the same flux unit (see figure below). It seems that the $R_{eco}$ magnitude with its daily cycle as well as the maximum GPP between two models are almost identical. However, urbanVPRM GEE starts to become negative way earlier than SMUrF GEE, leading to an earlier turning point for net biospheric uptake (~5 am in urbanVPRM vs. 6-7 am in SMUrF). We have now modified this sentence as follows:

> "In Boston, SMUrF reported similar NEE magnitude but with an hour delay where NEE becomes negative compared to urbanVPRM (Supplementary Fig. S9 vs. Figure 3B3 in Hardiman et al., 2017), likely due to discrepancies in the hourly data that drive two sets of hourly GEE fluxes."

[Figure]

*Figure S9. Monthly mean diurnal cycle of biogenic CO2 fluxes around the similar area considered in Hardiman et al. (2017).*

[Figure]

*Figure adopted from Hardiman et al., 2017*

*p.15 L12 "prediction" -> "predictions"*
*p.15 L13 "as" -> "of"*
*p.15 L19 "amount" -> "amounts"*
*p.16 L15 "comparison" -> "comparisons"*
*p.16 L15 "insights on" -> "insight into"*
*p.16 L25 "turns" -> "turn"*
*p.16 L26 "GPP," -> "GPP as well as"*

> All corrected.

*p.16 L31 Confusing sentence, please rewrite.*

> We have rewritten the relevant sentence: "Model discrepancies in producing $R_{eco}$ lead to an overall higher $R_{eco}$ and more positive NEE in SMUrF compared to urbanVPRM over LA (3rd column in **Fig. 11a**)."

*p.16 L32 "grids" -> "gridcells"*
*p.17 L11 "examine" -> "examined"*
*p.18 L25 "how" -> "how a"*
*p.18 L25 "bio-gradient" -> "gradient"*
*p.20 L6 "on-board" -> "onboard"*
*p.20 L28 "10" in "Q10" shouldn't be italicized*

> All corrected. We thank the referee #2 for pointing all these grammatical issues out.

---

## Author Response (AR1)

We thank both referees for their efforts and constructive comments. During the revision phrase, we tried our best to address the comments from two referees and modified figures, captions, and text throughout the paper to increase clarity and readability.

**Response to Referee #1**

We thank referee #1 for the constructive comments. Referee's comments are shown below in ***italics***, followed by our point-by-point responses in **blue** and changes in relevant text in **red**.

**Referee #1 Youngryel Ryu (Referee) ryuyr77@gmail.com**

*Dear Dien and all*

*I would like to congratulate you for this impressive manuscript. Incredibly comprehensive, in-depth analysis, great attentions to details and robust upscaling approach. I have to admit that I am not the expert in atmospheric transport model, where I didn't make any comment.*

*The authors developed a biogenic $CO_2$ balance model which includes GPP, $R_{eco}$, and NEE. They intended to develop this model for global cities, but actually it is applicable to the global land. The basic idea came from linking SIF and GPP. They developed the slopes between GPP and SIF (CSIF products) across FLUXNET sites. After fine tuning (e.g. crops) in the slopes, they converted CSIF (0.05 degree) to GPP. As urban landscape is heterogeneous, they used very high resolution land cover maps to apply the slopes for the relevant land cover types then aggregated to 0.05 degree. Then the authors developed an $R_{eco}$ model using NN with GPP, Tair and Tsoil. To evaluate the model performance, the authors compiled FLUXNET, INFLUX dataset and urbanVPRM model. Then the authors combined fossil fuel emissions data, $XCO_2$ data and an atmospheric transport model to tease out the contributions of biogenic $CO_2$ fluxes in urban CO2 fluxes around the world.*

*The scope of this manuscript is vast but the authors didn't gloss over important details. Although some parts could be improved further, overall I see this is already too good. Though I would like to make some suggestions for further improvement.*

We truly appreciate the recognition and constructive comments from referee #1 Youngryel Ryu and have tried our best to conduct additional analyses and to improve the presentation of this manuscript.

*First, evaluate SMUrF NEE directly against FLUXNET data like what you did for GPP and $R_{eco}$ in Fig 5. Good performance in GPP and $R_{eco}$ does not necessarily indicate good performance in NEE which is tiny signal compared to the other two fluxes. The authors reported Fig S10 for NEE evaluation, but I feel it is not enough. It is fine to report rather poor performance in NEE, which is quite well expected as machine learning based NEE (e.g. FLUXCOM) performed poorer than GPP and $R_{eco}$. It would be an useful point about how to improve SMUrF later.*

We agree that NEE evaluation against observations is critical. In the initial manuscript, we have already included a model evaluation of the HOURLY mean biome-specific NEE fluxes against FLUXNET data (Fig. 9 and Sect. 3.2.1). In accordance with the reviewer's point, NEE has a poorer performance compared to GPP and $R_{eco}$, especially for biomes with less training data (i.e., FLUXNET observations).

*Second, the current evaluation focused on diurnal to seasonal scales. Could you provide some discussion on the model performance in interannual to trends? e.g. in case of LA, how NEE varied across dry and wet years? How does NEE/fossil fuel $CO_2$ varies across dry and wet years?*

While the paper would be more informative with additional discussion on the interannual variation/trend in urban NEE, the primary motivation of developing the model is to separate the anthropogenic and biogenic $CO_2$ signals when interpreting the atmospheric measurements. The column $CO_2$ retrieved from satellites may be affected by the upstream fluxes over timescales of only O(day). Therefore, we argue that $CO_2$ fluxes at the diurnal scale matters the most to the downwind atmospheric $CO_2$ observations than $CO_2$ fluxes at much longer timescales. For example, the interannual variation/trend in NEE may be a secondary-order effect, superimposed on the urban-rural difference in NEE fluxes that influence the $CO_2$ background determination. Finally, the paper is already quite lengthy, and considering the main motivation of the manuscript, we are inclined not to touch on NEE fluctuation at the moment, but added relevant text in the future work section for clarification (Sect. 4.2, P20L14-L19):

"Because atmospheric $CO_2$ concentrations measured from satellites are mainly influenced by the anthropogenic and biogenic carbon fluxes a few hours to days ahead of the overpass time, this work focused on presenting and evaluating the diurnal and seasonal $CO_2$ fluxes. Biogenic $CO_2$ fluxes at other moments, e.g., their interannual variations and trend, may require further investigations. We hope to examine more cities and different times of the day in future

studies to better quantify the relative biogenic and anthropogenic contributions to $XCO_2$ anomalies. Incorporating uncertainties in biogenic fluxes and resultant $XCO_{2.bio}$ is needed for future top-down studies with aims of quantifying urban signals especially over growing seasons."

*Third, I would like to recommend adding some discussions for including evaporation in SMUrF, not now but in v2. Your model already has most important components to compute evaporation. One approach would be to use Ball-Berry model to link your GPP, canopy conductance and finally evaporation. I really enjoyed this paper (https://doi.org/10.1073/pnas.2005253117), which stressed the important linkage between irrigation and biogenic $CO_2$ fluxes in LA. I think SMUrF can track this as well once evaporation module is included.*

We thank the reviewer for the suggestion and agree that an evaporation module would be a great function to be implemented in the future. Urban irrigation plays an important role in GPP especially for semiarid urban areas, (Johnson and Belitz 2012, Vahmani and Hogue 2014, Miller et al., 2021). We added the following discussions in **Sect. 4.2 (now on P20 L6-L12)**:

"More challenging is the shortcoming that current flux estimates in SMUrF over cities still rely on relationships derived from observations over natural biomes. Urban trees are found to possess different characteristics from natural trees (Smith et al., 2019), which pose a difficult task for biospheric models without more dedicated observations and mechanistic understanding of the urban environment. Anthropogenic moisture input (i.e., urban irrigation) has been found to effectively influence urban biogenic fluxes, particularly over (semi)arid residential areas (Johnson and Belitz, 2012; Vahmani and Hogue, 2014; Miller et al., 2021). Although we currently rely on SIF to pick up possible irrigation effect on GPP, it would be interesting to explore the linkage between water and carbon fluxes in future analyses."

**Followings include minor comments:**

*P4 L5-10: The previous paragraph criticized the limitation of simple $R_{eco}$ model, then this paragraph explained ML for SIF and land surface fluxes. I feel somewhat disconnected from the previous paragraph.*

We thank the reviewer for pointing out the disconnection. Our initial thought is to highlight the difficulties and limits in estimating $R_{eco}$, which is also the motivation of adopting ML technique for $R_{eco}$ estimates. We have now improved the flow in between the text (**now on P4 L4 – L8**):

"...After all, the complexity of biological and non-biological processes of $R_{eco}$ and the lack of mechanistic understanding of how biotic and abiotic factors affect $R_{eco}$ render challenging mechanistic modeling of $R_{eco}$. Given the complexity in modeling $R_{eco}$, we will turn instead to ML techniques that have been increasingly applied in many disciplines to help answer complicated, entangled problems via extracting patterns from data streams for predictions and generalizations...."

*P9: pure temperature -> revise*

We have now changed to "direct temperatures and GPP observations from EC towers" (now on P9 L10).

*P10 L30: I feel the assumption for no correlation between GPP and $R_{eco}$ is overly simplified. SMUrF model structure indicates GPP is a forcing to $R_{eco}$ (P6 L16).*

We agree that the assumption for no correlation is overly simplified. After double checking the uncertainties provided in the output netcdf4 files, we confirmed that no uncertainties associated to hourly NEE is provided. Calculating NEE uncertainties with the correlation between GPP and $R_{eco}$ will involve regenerating all model fields, we may skip the NEE uncertainties for now. However, we do provide the individual grid-level uncertainties to the GPP and $R_{eco}$ files. We apologize for the incorrect statement in the previous version and clarified the text (**now on P10 L18**):

"Here we estimated errors in modeled GPP and $R_{eco}$ based on FLUXNET observations."

*P13 L6: What's GEE? Isn't it GPP?*

GEE is the gross ecosystem exchange and closely related to GPP. GEE is often used by researchers working with eddy covariance observations, while GPP is used more often by ecologists. We define GEE = -GPP, so that photosynthetic uptake represents removal from the atmosphere (Fig. 6). We modified the relevant text (now on P12 L22).

*P18 L12: what is QF?*

QF stands for quality flag, a measure provided by the OCO-2 XCO$_2$ retrieval to indicate data quality. We have now modified that sentence (now on P17 L14).

*P20 L3: spatial SIF -> revise*

We have now changed to "SIF retrieval with a broader spatial coverage over cities" (now on P19 L26).

*P20 L10-22: It is worth discussing complex SIF-GPP relationships reported in recent literature. Consistent, linear relationship disappears in some cases. https://doi.org/10.1016/j.rse.2018.07.008 https://doi.org/10.1002/2017JG004180*
*https://doi.org/10.1038/s41598-018-32602-z*

We thank the reviewer for sharing these relevant publications and have added relevant text on the nonlinearity of GPP-SIF relationship in the methodology section (Sect. 2.2, now on P7 L25 – 32):

"While non-linear relationships between SIF and GPP at leaf- and canopy- level have been observed (Helm et al., 2020; Magney et al., 2017; Maguire et al., 2020; Marrs et al., 2020; Verma et al., 2017), GPP is observed to be linearly related to SIF at increasing temporal and spatial (ecosystem and regional) scales (Frankenberg et al., 2011; Sun et al., 2017) as leaf-level differences in composition, light exposure, stress, and stress response mix out (Magney et al., 2020). Considering uncertainty in CSIF and flux tower-partitioned GPP as well as the noise in the GPP-SIF relationship across global flux sites (**Supplementary Fig. S1**), we adopted linear fits instead of non-linear fits between GPP and CSIF. Errors due to departure from linearity will be implicitly included in GPP uncertainties calculated from model-tower validations (**Sect. 2.5**)."

as well as the discussion section (Sect. 4.2, now on P19 L6 - 19):

"The adoption of SIF has dramatically benefited the GPP calculation over urban areas around the globe, as non-vegetated surfaces within the satellite footprint do not contribute to observed signals. However, the main caveat lies in the assumption of linear GPP-SIF relationship and one set of constant $\alpha$ values across all seasons used in SMUrFv1. Previous research (Magney et al., 2020; Miao et al., 2018; Wohlfahrt et al., 2018; Yang et al., 2018) revealed divergence of the empirical linear GPP-SIF relation at sub-diurnal and leaf scales, and under certain environmental conditions (low light, or high light & stress), owing to competing fluorescence, photochemical, and non-photochemical pathways for the absorbed light (Magney et al., 2020). For example, Yang et al. (2018) suggest considering additional environmental and biophysical factors related to the modeling of light use efficiency, e.g., relative humidity, cloudiness, and growth stage of crops, to improve SIF-based GPP estimates. Although multiple studies have shown dependence of the linear slope on PFT (e.g., Guanter et al., 2012; Sun et al., 2017; Turner et al., 2021), further research is needed to understand the scale dependence of the GPP-SIF relation, and determine if an inflection point for linearity exists. Given noise/uncertainty in the CSIF product and EC tower data across multiple continents, we apply a simple linear regression fit and let the uncertainty analysis incorporate deviations from the linear assumption. Future iterations of SMUrF can test alternative statistical fits with physical fundamentals or expand the GPP-SIF slopes across seasons, and new urban land cover maps (e.g., Coleman et al., 2020)."

*Again, this is a great manuscript. I really enjoyed reading it, and also learned a lot.*
*Thanks- Youngryel*

Thank you, Youngryel. – Dien on behalf of the team

**Response to Referee #2**

We thank referee #2 for the constructive comments. Referee's comments are shown below in *italics*, followed by our point-by-point responses in **blue** and changes in relevant text in **red**.

**Anonymous Referee #2**

*This paper proposes a new model for estimating biogenic carbon fluxes from urban areas. This model, called SMUrF uses a new global solar-induced fluorescence product (cSIF) and biome specific GPP-SIF relationships to create a temporally and spatially explicit flux product specifically turned for urban vegetation, which is notoriously difficult to model accurately. Respiration is also carefully considered in this new model, using a neural net approach. The paper does an excellent job of describing the intricate (and very numerous) processes involved and the model, and the result is a truly exciting work that is sure to be of great interest to the flux modelling community. Although the SMUrF model contains a large number of assumptions (as any model of this scope does), and will likely be refined in the future, the authors cleverly acknowledge that this version Discussion paper is only the first iteration of the SMUrF model by referring to this version as "v1" in the main title.*

We appreciate the constructive feedback from the reviewer and have tried our best to improve the clarity of the text plus figures and redo the demonstration on analyzing column $CO_2$ observations.

*The model relies on an assumption that GPP and SIF have a linear relationship, and that the slope of this relationship (alpha) is only a function of biome type. The plots for these calculations are buried in the supplemental, and for many of the biomes, the relationship does not appear to be linear. It is unclear if this non-linearity in addressed in the uncertainty analysis. This is the one part of the analysis that I wish was discussed in more detail.*

We are aware of the non-linearity between GPP and SIF at finer temporal scales (e.g., sub-diurnal scales) and added two discussions on the nonlinearity of GPP-SIF relationship in the methodology section (**Sect. 2.2, now on P7 L25 − 32**):

"While non-linear relationships between SIF and GPP at leaf- and canopy- level have been observed (Helm et al., 2020; Magney et al., 2017; Maguire et al., 2020; Marrs et al., 2020; Verma et al., 2017), GPP is observed to be linearly related to SIF at increasing temporal and spatial (ecosystem and regional) scales (Frankenberg et al., 2011; Sun et al., 2017) as leaf-level differences in composition, light exposure, stress, and stress response mix out (Magney et al., 2020). Considering uncertainty in CSIF and flux tower-partitioned GPP as well as the noise in the GPP-SIF relationship across global flux sites (**Supplementary Fig. S1**), we adopted linear fits instead of non-linear fits between GPP and CSIF. Errors due to departure from linearity will be implicitly included in GPP uncertainties calculated from model-tower validations (**Sect. 2.5**)."

as well as the discussion section (**Sect. 4.2, now on P19 L6 - 19**):

"The adoption of SIF has dramatically benefited the GPP calculation over urban areas around the globe, as non-vegetated surfaces within the satellite footprint do not contribute to observed signals. However, the main caveat lies in the assumption of linear GPP-SIF relationship and one set of constant $\alpha$ values across all seasons used in SMUrFv1. Previous research (Magney et al., 2020; Miao et al., 2018; Wohlfahrt et al., 2018; Yang et al., 2018) revealed divergence of the empirical linear GPP-SIF relation at sub-diurnal and leaf scales, and under certain environmental conditions (low light, or high light & stress), owing to competing fluorescence, photochemical, and non-photochemical pathways for the absorbed light (Magney et al., 2020). For example, Yang et al. (2018) suggest considering additional environmental and biophysical factors related to the modeling of light use efficiency, e.g., relative humidity, cloudiness, and growth stage of crops, to improve SIF-based GPP estimates. Although multiple studies have shown dependence of the linear slope on PFT (e.g., Guanter et al., 2012; Sun et al., 2017; Turner et al., 2021), further research is needed to understand the scale dependence of the GPP-SIF relation, and determine if an inflection point for linearity exists. Given noise/uncertainty in the CSIF product and EC tower data across multiple continents, we apply a simple linear regression fit and let the uncertainty analysis incorporate deviations from the linear assumption. Future iterations of SMUrF can test alternative statistical fits with physical fundamentals or expand the GPP-SIF slopes across seasons, and new urban land cover maps (e.g., Coleman et al., 2020)."

*In an effort to relate the SMUrF model output to XCO$_2$ observations from OCO-2, the X-STILT transport model was used to generate total-column footprints. The assemblage of Boston area footprints shown in Figure 12a. shows a satellite overpass that occurred while the winds were out of the NNE along the flight track. The forward model results shown in Figure 12b seem ok, but*

*the plots in 12b,c are confusing, because they are not XCO$_2$, they are the spatially explicit contributions to the XCO$_2$ concentrations for the satellite observations.*

Correct - Figure 12 c,d show the anthropogenic and biogenic contributions in ppm from each upwind gridcell with respect to downwind receptors. As mentioned in the figure caption, we referred to those anthropogenic and biogenic contributions as spatial XCO$_{2.ff}$ and XCO$_{2.bio}$. These spatial contributions have units of ppm and are further calculated from the product between X-STILT column footprint [ppm /($\mu$mol m$^{-2}$ s$^{-1}$)] and upwind fluxes [$\mu$mol m$^{-2}$ s$^{-1}$]. The spatial sum of these contributions arrives at the total anthropogenic and biogenic anomalies at corresponding receptors, as shown in panel b). We have now modified relevant text for clarification (**now in Sect. 3.3 on P16 L19 – L22**).

*The analysis in Figure 12e seems problematic, particularly for the treatment of the background concentration. The background value chosen appears somewhat arbitrary and taken from a region downwind of the city. The correlation between the binned OCO-3 observations (black triangles) and the full model result (purple line) is not particularly strong. The author states that the additional of SMUrF to the analysis is an improvement over just using a fossil fuel inventory, but other papers (such as the cited Sargent, 2018) spend a lot more time dealing with incorporating the biosphere with these types of transport models. While SMUrF represents an important step forward in assimilating SIF measurements into a biosphere carbon flux model, the STILT analysis at the end is incomplete, and, in my opinion, the paper would be better off for dropping this part entirely. Many researchers will surely be eager to explore the use of SMUrF with transport models to compare with satellite data, but these comparisons will need to spend a lot more time on dealing with subtleties such as determining the background. Because XCO$_2$ anomalies are so small over cities (typically a few ppm at most), a careful error analysis would also be needed, which is lacking here.*

1) We agree with the reviewer's criticism - it might be too soon to conclude that using SMUrF can effectively improve the background along satellite swath, particularly given only one case examined. However, we hope to provide a demonstration of how SMUrF can be used with transport models and emphasis the role of urban-rural gradient in NEE and resultant biogenic XCO$_2$ signals played in the background definition. We feel that even though this is a model description paper, it is illustrative for the reader to see an application of the model in helping to interpret satellite XCO$_2$ data.

**To deemphasize the quantitative results from a limited number of analyses, we modified a relevant sentence in the Abstract (now on P1 L27-28) to read.**

Initial text - "By examining a few summertime satellite tracks over four cities, we found that the urban-rural gradient in column CO$_2$ (XCO$_2$) anomalies due to NEE can sometimes reach ~0.5 ppm and be close to XCO$_2$ enhancements due to FFCO$_2$ emissions."

Modified text - "To illustrate the application of SMUrF, we used it to interpret a few summertime satellite tracks over four cities and compared the urban-rural gradient in column CO$_2$ (XCO$_2$) anomalies due to NEE against XCO$_2$ enhancements due to FFCO$_2$ emissions."

[Figure]

2) In terms of the background definition, we agree that the initial choice can be arbitrary and calculating the proper background for column data is especially challenging and has been extensively investigated in our previous work (Wu et al., 2018, hereinafter Wu2018). **We now followed the overpass-specific approach illustrated in Wu2018, that is calculated from the average observed XCO$_2$ over the background latitude band.** The relevant figure regarding the overpass-specific approach has been added to the supplement (**now as Supplementary Fig. S13**) in the revised manuscript.

Specifically, we leveraged the forward-mode of STILT where STILT particles were continuously released forward in time from a box around Boston for 12 hours. The border of the urban plume is defined by fitting a normalized 2D kernel density (purple contour in the **Fig. S13** to the right) to particle locations during the few minutes where OCO-2 overpass the city. The intersection of the urban plume and the satellite swath give rise to the urban-polluted latitude range (red triangles in panel b). Then the background latitude range (42.26 to 42.76°N, green ribbon in panel b) is chosen to the north of the urban-polluted range, given 1) the geometry

between the swath orientation and the wind direction and 2) possible contamination from oceanic fluxes over the region to the south of the urban plume.

As a result, we arrived at both the mean background along with its uncertainty for this swath as shown in dotted-dashed green line and green ribbon, which have values of 403.37 +/- 1.03 ppm. The background uncertainty can also be used in error analyses and atmospheric inversions like those conducted in Wu2018.

3) **Next, we explain the reason for using this overpass-specific background**:

Wu2018 has carefully invested three common methods with different complexity in estimating $XCO_{2.bg}$. The reviewer is welcomed to read over Sect. 2.3 and Sect. 3.3 in Wu2018 for full details. For the convenience, we summarized main messages as follows and adopted the relevant two figures from Wu2018 (Figure S8 and S12, also shown on the next page) for explanations:

M1. a "trajectory-endpoint" method is investigated by assigning $CO_2$ values extracted from global models (e.g., CarbonTracker, CT) to trajectory endpoints including simulating biospheric, oceanic, and prior components. This method has been widely used in Lagrangian-based modeling work including the cited Sargent et al., 2018. **However, most prior work only had to deal with $CO_2$ measurements within the PBL where huge $CO_2$ anomalies are caused by either anthropogenic or biogenic. When applying this trajectory-endpoint method solely relying on model simulations to interpreting column $CO_2$ measurements, this approach may often lead to potential "bias" in background values (orange lines in Figure S12 adopted from Wu2018).** Although there is hardly a "truth" for $XCO_2$ background, the modeled total $XCO_2$ based on this approach appears to be unreasonably higher or lower than the retrieved $XCO_2$ by 1-2 ppm (orange dotted-dashed line vs. black solid line in Figure S8 of Wu2018). These mismatches of 1-2 ppm can already be huge given small anthropogenic $XCO_2$ enhancements and is caused by potential uncertainties in the adopted global models (e.g., CT) with accumulated errors in the endpoint of STILT (further result from wind errors).

M3. an "overpass-specific" background as described earlier. **$XCO_{2.bg}$ calculated by combining observations and wind information from forward-mode of STILT can be more consistent with the retrieval and account for upwind-downwind geometry, which is better than approaches that solely rely on models OR observations. The biggest hurdle would be wind bias in STILT, which unfortunately can affect the M1 trajectory-endpoint background as well.**

*The following Figure S12 and Figure S8 are adopted from Wu et al. (2018).*

[Figure]

**Figure S12.** Same as **Fig. 6e**, except for using OCO-2 Lite b8. Numbers labeled in darkgreen donote the amount of screened soundings (QF = 0) using b8 in the background. Due to only 8 soundings for overpass on 2014122910, background uncertainty is hard to estimate (no error bar displayed).

[Figure]

**Figure S8.** Same as **Fig. 8**, but for all five overpasses examined over Riyadh using OCO-2 Lite v7.

4) Lastly, we agree with the reviewer that the biogenic adjusted signal may not correlate strongly with the observations, possibly due to various reasons, e.g., 1) the bias in near-field wind direction, 2) uncertainties in both FF and biogenic fluxes, 3) retrieval error. We have now added a wind error analysis by comparing the modeled wind speed and directions against a NOAA radiosonde station (41.67N, 69.97W) adjacent to Boston city. Close to the overpass hour (07/07/2018 17 UTC), we see overall positive biases in the HRRR-based wind direction from the surface to 3 km. This positive bias potentially explains a northward modeled $XCO_2$ peak than the observed peak (latitude shift of about 0.1 degreeN in **Fig. 12e**).

| time.string | u.bias | v.bias | ws.bias | wd.bias | rmse |
|---|---|---|---|---|---|
| 07/07/2018 00 UTC | -2.29 | -1.86 | -1.36 | 16.6 | 3.08 |
| 07/07/2018 12 UTC | 0.220 | 1.22 | -1.70 | 2.68 | 1.91 |

Revised Figure 12e:

[Figure]

We may argue that if the aim of a study is to quantify FF emissions over cities surrounded by vegetations (e.g., Sargent et al., 2018), a comprehensive error analysis or even an atmospheric inversion is needed. In future work, we will follow the full error analysis and potentially a scaling factor type atmospheric inversion conducted in Wu2018 to make more quantitative results. However, given the already lengthy manuscript and the main scope of this work being model presentation, we simply modified the text in Sect 4.1 (now on P18 L16 – L31) for clarifications.

> "To facilitate visualization and understanding of $\Delta XCO_{2.bio}$ and bio-adjusted background, let us return to the Boston case (**Fig. 12**). Following the "overpass-specific approach" proposed in Wu et al. (2018), we estimated the urban plume (black curve in **Supplementary Fig. S13a**) and defined the background latitude range of 42.26°– 42.76° N (light green ribbon in **Fig. 12e**). The constant background is 403.37 ppm (dark green line in **Fig. 12e**) with an uncertainty of 1.03 ppm containing both retrieval errors and observational noise (**Supplementary Fig. S13b**). The mean $XCO_{2.ff}$ and $XCO_{2.bio}$ anomalies within the background region are 0.23 ppm and –1.41 ppm, respectively. After integrating the bio-gradient $\Delta XCO_{2.bio}$, a new bio-adjusted background varies along latitude (light green line in **Fig. 12e**). If modeled $XCO_{2.ff}$ is added to the bio-adjusted background, the resultant total $XCO_2$ better reproduces the latitudinal variations of the measured mean values (**Fig. 12e**). Both the observed $XCO_2$ and modeled $XCO_2$ with the $\Delta XCO_{2.bio}$ correction term exhibit dips in $XCO_2$ on both sides outside the urban peak, which is missing from the model result using the constant background (orange line in **Fig. 12e**).
>
> A more comprehensive error analysis of various modeled and observed errors is needed to draw further quantitative conclusions from the model-data $XCO_2$ comparison. For instance, modeled $XCO_2$ appears to extend wider latitudinally with a lower amplitude and a small latitude shift of ~0.1° compared to observed $XCO_2$ (purple line versus black triangles in **Fig 12e**) likely due to bias in wind speed and direction. Nonetheless, neglecting the latitudinal/spatial gradient in biogenic $XCO_2$ anomalies given gradients in NEE can affect the extracted urban signal and inferred $FFCO_2$ emissions."

We also reemphasis the future needs in Sect. 4.2 and deemphasis the quantitative conclusion in the abstract (as mentioned in above point 1):

> "We hope to examine more cities and different times of the day in future studies to better quantify the relative biogenic and anthropogenic contributions to $XCO_2$ anomalies. Incorporating uncertainties in biogenic fluxes and resultant $XCO_{2.bio}$ is needed for future top-down studies with aims of quantifying urban signals especially over growing seasons."

In summary, we clarify the key point in this manuscript being the urban-rural gradient in biogenic fluxes and $CO_2$ anomalies. Even though one may not follow the same exact constant background approach with biogenic adjustments as we showed in Sect. 4.1, one needs to consider the urban-rural contrast in biogenic fluxes that is lacking in many $XCO_2$-based studies.

*The manuscript contains a large number of figures, many with numerous subplots. While this isn't uncommon for GMD papers describing a new model, this particular work would benefit from slimming down some of the figures. I've discussed a few of the figures individually below:*

> We appreciate these individual comments and have made some rearrangements to the figures.

*Figure 1: This is a really well laid out flow chart. It took me a while to get through it all, but it was really helpful in understanding the model, and I like how it was labeled with section and figure references.*

> We thank the reviewer for the recognition and are very glad this flow chart worked well in the end.

*Figure 2: Subplot c needs units for alpha values. Also, subplots are not labeled.*

> We have now added the unit for $\alpha$ values in the figure caption, i.e., (umol m$^{-2}$ s$^{-1}$) / (mW m$^{-2}$ nm$^{-1}$ sr$^{-1}$).

*Figure 6: This figure is way too complicated. In addition to their being too many cities, I can't easily discern what the take home message is supposed to be from all of these plots.*

We may argue that this zoom-in-and-out panel plot gives a nice overview of 1) anthropogenic and biogenic $CO_2$ fluxes from urban center to its surrounding and 2) how regional total $CO_2$ fluxes vary with seasons. One can spot the urban hotspots where anthropogenic $CO_2$ "beat" down the biogenic $CO_2$ fluxes.

*Figure 7: Again, too many subplots. It would be easier to read if thier were fewer cities selected. To me, the interesting information in this figure is both the magnitude of max NEE for different cities and the timing of when that max NEE occurs. Perhaps it would be more impactful to show a different type of plot. Perhaps a scatter plot with the x-axis being day-of-year for the NEE peak and the y-axis being peak magnitude? You could then pack a bunch more cities into one plot, and label the cities in the scatterplot.*

We appreciate the suggestions from the reviewer but may argue that the current presentation can provide a broad view for cities across multiple continents and hopefully facilitate readers with different cities of interests.

*Figure 8: These time of day plots are nice, but the half-circle makers are hard to see.*

We have now replaced circles with solid dots.

*Figure 10: Again, too many panels.*

We have now moved two of the initial six panels to the supplement (now as Fig. S12).

*Figure 11: This is great. I wish there was more urbanVPRM comparisons with other cities. A real test of the usefulness of SMUrF is its performance compared to other models, especially those also tailored for urban areas.*

Yes – we also wish to provide more model comparisons with other observations and model products for more locations, which require additional support and collaboration from data providers/users in the field.

*Overall, this is an impressive manuscript. The model described is sure to make an impact in the community, and I know that I and other researchers look forward to working with it.*

We thank the support from the reviewer and will keep improving the SMUrF model, given increasing understanding towards SIF, respiration, and urban biosphere as well as the availability of upcoming remote sensing data.

**Specific line-by-line comments (mostly grammar stuff) are below:**

*p.6 L3 Grammar (". . .by trained on. . .")*

Corrected – changed 'by trained on' to 'that trained on'

*p.6 L10 "Laser" capitalized*

Corrected – "Laser" to "laser".

*p.6 L12 punctuation*

We've modified this sentence as "AGB and its grid-level uncertainty [tons ha$^{-1}$] by definition describe the "oven-dry weight of the...".

*p.6 L28 Underline on part of "(Sect. 3.2)."*
*p.13 L34 "than" -> "rather than"*

Corrected.

*p.14 L26 Why? Please add a sentence of explanation.*

The point where NEE becomes negative is the number we read from Figure 3A3 and 3B3 in Hardiman et al. 2017 (also attached to the right). They show that the NEE turned negative at ~5 am local time in July 2013. We extract SMUrF fluxes from the similar Boston area considered in Hardiman2017 and calculated the monthly mean diurnal cycle with the same flux unit (see figure on the next page, now as Supplementary figure S9). It seems that the $R_{eco}$ magnitude with its daily cycle as well as the maximum GPP between two models are almost identical.

[Figure]

**Fig. 3.** Cumulative C fluxes (MgC ha$^{-1}$) for 2013 indicate that MA is a net biogenic C sink and that on an area-basis statewide anthropogenic emissions are of similar magnitude as biogenic fluxes (A1). Both biogenic and anthropogenic fluxes (kgC ha$^{-1}$ h$^{-1}$) follow daily cycles, the amplitude of which varies seasonally (A2, A3). Boston's biogenic fluxes are dwarfed by anthropogenic C emissions (B1) but follow similar patterns as seen at the state-level (B1, B2). Anthropogenic fluxes in all B panels are plotted on a second y-axis to facilitate comparison with biogenic fluxes.

*Figure adopted from Hardiman et al., 2017*

However, urbanVPRM GEE starts to become negative way earlier than SMUrF GEE, leading to an earlier turning point for net biospheric uptake (~5 am in urbanVPRM vs. 6-7 am in SMUrF). We have now modified this sentence (**now on P14 L12-L14**) as follows:

[Figure]

**Figure S9. Monthly mean diurnal cycle of biogenic CO2 fluxes around the similar area considered in Hardiman et al. (2017).**

"As for Boston, SMUrF reported similar NEE magnitude compared to urbanVPRM but with an hour delay where NEE becomes negative (**Supplementary Fig. S9** vs. Figure 3B3 in Hardiman et al., 2017), likely due to discrepancies in the hourly data that drive the hourly GPP fluxes."

p.15 L12 "prediction" -> "predictions"
p.15 L13 "as" -> "of"
p.15 L19 "amount" -> "amounts"
p.16 L15 "comparison" -> "comparisons"
p.16 L15 "insights on" -> "insight into"
p.16 L25 "turns" -> "turn"
p.16 L26 "GPP," -> "GPP as well as"

All corrected.

p.16 L31 Confusing sentence, please rewrite.

We have rewritten the relevant sentence (now on P16 L2-L4)

"An overall higher $R_{eco}$ and more positive NEE is associated with SMUrF compared to urbanVPRM over LA (3rd column in **Fig. 11a**), attributed to methodological discrepancies in producing $R_{eco}$."

p.16 L32 "grids" -> "gridcells"
p.17 L11 "examine" -> "examined"
p.18 L25 "how" -> "how a"
p.18 L25 "bio-gradient" -> "gradient"
p.20 L6 "on-board" -> "onboard"
p.20 L28 "10" in "Q10" shouldn't be italicized

All corrected. We thank the referee #2 for pointing all these grammatical issues out.